# PBAF/cBAF reorganization on H3.3 chromatin regulates BMAL1 activity in the absence of circadian negative feedback

**Dominika Letkova** [1], **Aurelie Peticca** [1], **Damien Sery**[1], **Masahide Seki** [2], **Yutaka Suzuki** [2] **& Kiran Padmanabhan** [1,3] ✉

Circadian rhythms in gene expression are coincident with 24 hr dynamics in the recruitment of the core-clock transcription factor heterodimer CLOCK-BMAL1 to chromatin. In the liver, circadian chromatin is characterized by rhythmic histone modifications and the deposition of histone variant H2A.Z. However, other histone variants and remodelers that work in conjunction with CLOCK-BMAL1 on variant chromatin remain poorly understood. Here, we reveal that H3.3 variant histone deposition peaks during the daytime in liver chromatin and that CLOCK-BMAL1 is recruited to H3.3 nucleosomes. Moreover, H3.3:CLOCK-BMAL1 associates with PBAF and BRG1/cBAF complexes - members of the SWI/SNF remodeler family - only during the active phase of the circadian cycle. In clock-disrupted *Per1*[-/-]; *Per2*[-/-] livers, we observe a depletion in ARID2, the central cog in the molecular assembly of the PBAF complex, accompanied by an increase in H3.3 incorporation. Remarkably, a disassembly of PBAF complex and the concurrent reduction in BRG1 triggers a remodeler reorganization in *Per* knockout livers, where BRM/cBAF now targets BMAL1 at readily-accessible genomic sites. An abundance of fragile, acetylated H3.3 nucleosomes and a remodeler reorganization provide a mechanistic basis for BMAL1 activity in the absence of PER-mediated negative feedback.

Circadian rhythms orchestrate organismal physiology and behavior to anticipate daily changes in the environment. At the cellular level, mammalian circadian rhythms are built upon a transcriptional-translational feedback loop (TTFL), wherein the activators BMAL1 and CLOCK (and its paralog NPAS2) are repressed by their transcriptional output- the period (PER1, PER2, and PER3) and cryptochrome (CRY1 and CRY2) proteins[1].

The chromatin environment, shaped by the wrapping of DNA around nucleosomes, regulates the accessibility of transcription factors (TFs) to their genomic target sites. CLOCK-BMAL1, a pioneer transcription factor, can bind to E-box sequences within nucleosomes located in facultative heterochromatin[2,3]. This interaction facilitates nucleosome eviction, leading to the formation of nucleosome-depleted regions[2]. Post-translational modifications (PTMs) on histone tails, along with the

specific composition of the histone octamer, further influence chromatin states and transcription factor accessibility. Rhythmic cycles of histone acetylation, deacetylation, methylation, and ubiquitination at E-boxes and flanking regions allow for 24-hour cycles in open and closed chromatin states[4,5]. Many histone and chromatin-modifying complexes responsible for these modifications are themselves recruited either by CLOCK-BMAL1 or by repressive PER complexes in mammals[6]. In other species, such as *Neurospora crassa*, dynamic remodeling of nucleosomes occurs upon WCC binding at C-box sequences near *frq* promoter. The recruitment of SWI/SNF remodelers by WC-1 results in nucleosomal eviction and chromatin looping at these sites, followed by transcriptional activation of *frq* gene and activation of the circadian cycle[7]. In *Drosophila*, the BRAHMA (BRM) remodeler tempers CLOCK-mediated transcriptional outputs by recruiting repressive factors and generating

[1]Institut de Génomique Fonctionnelle de Lyon, Univ Lyon, CNRS UMR 5242, Ecole Normale Supérieure de Lyon, Université Claude Bernard Lyon, Lyon, France. [2]Graduate School of Frontier Sciences, The University of Tokyo, Chiba, Japan. [3]INSERM, Paris, France. ✉e-mail: kiran.padmanabhan@ens-lyon.fr

nucleosome occupied regions[8]. In mammals, rhythmic recruitment of SRC-2/PBAF SWI/SNF remodeler complexes has been shown to facilitate REV-ERB alpha loading on chromatin. Subsequent dynamics in RNA Polymerase II recruitment, and the regulation of steps involved in productive elongation and termination generate stable circadian rhythms in mRNA expression[4,9,10].

Beyond PTMs, chromatin states are also modulated by the exchange of canonical histones with histone variants. The H2A variant, H2A.Z can regulate transcription either by making direct contacts with the transcription machinery or by acting as a binding platform for pioneer transcription factors[11]. 24 hr cycles in H2A.Z deposition on chromatin have been described in mammals and other species including *Arabidopsis*, *Neurospora* and implied in *Drosophila* clock neurons[2,12–14]. While H2A.Z is required for BMAL1 binding and CLOCK-BMAL1 activity, its progressive deposition on chromatin over the subjective night and resulting compaction ensures PER-mediated circadian feedback[15].

The deposition of the H3 histone variant, H3.3, at genes is concomitant with their transcriptional activation across many model systems[16,17]. In many cases, this process involves the formation of fragile nucleosomes containing both H3.3 and H2A.Z, within the same histone octamer[18–20]. Post-translational modifications further enhance the dynamics of these variant nucleosomes. Acetylation of the amino-terminal tail of H2A.Z as well as at H3.3 lysine residues K115 and K122 (H3.3K115ac and H3.3K122ac) on the lateral surface of the histone octamer increases the mobility of these variant nucleosomes, facilitating their role in transcriptional regulation[11,21,22].

The compaction of chromatin at night is coincident with the full assembly of the repressive NuRD remodeler complex at CLOCK-BMAL1 binding sites[23]. By contrast, during the day, CLOCK-BMAL1 interacts with CHD4, a specific subunit of the NuRD complex to promote circadian gene expression[23]. In dividing cells, the NuRD complex has been shown to assemble on histone variant H3.3 marked chromatin to activate transcription[24]. Additionally, the SWI/SNF remodeler subunit ARID1A is required for recruiting CHD4 to H3.3 nucleosomes[25]. However, it remains unclear whether similar mechanisms operate in tissues. Could 24 hr cycles in fragile nucleosomes provide a chromatin platform for CLOCK-BMAL1-mediated transcriptional activation and/or establish PER-mediated repression? To address these questions, we investigated the role of the H3.3 variant and associated chromatin remodelers in the establishment and function of the mammalian circadian clock.

## Results

To gain insight into H3.3 chromatin states over circadian time, we analyzed H3.3 nucleosome complexes in wildtype liver chromatin at time points corresponding to the peak and trough of CLOCK-BMAL1 activity. In parallel, we also analyzed H3.3 complexes in *Per1*[−/−]; *Per2*[−/−] knockout (PerKO) liver tissue, where BMAL1 activity is high in the absence of circadian negative feedback[26,27].

Since H3.3 differs from canonical H3.2/1 by only 4-5 amino acids, we used a conditional knock-in mouse model wherein a 2X-FLAG-HA epitope sequence was inserted in frame with the N-terminus of H3.3A to purify variant complexes[28]. Heterozygous FH-H3.3A mice, carrying a single copy of the epitope-tagged H3.3A protein in a wildtype or a PerKO background were then raised under controlled conditions and sacrificed during the subjective day and night for isolation of the liver and preparation of chromatin. Soluble chromatin enriched for mononucleosomes was subjected to native HA-tag immunoprecipitation (IP) and tested for the presence of the core-clock transcription factors. H3.3A complexes retained both CLOCK and BMAL1 at CT8 while they were absent during the repressive clock phase at CT20 (Fig. 1a). Remarkably, H3.3A complexes retained H2A.Z but did not include canonical H3.1/2 at the two time points. Strikingly, in PerKO

animals CLOCK-BMAL1 was no longer enriched on variant H3.3 nucleosomes.

We then scaled up H3.3A complex purification from liver nuclei to identify H3.3A-associated proteins. Soluble chromatin enriched for mononucleosomes, from day and night-time time points and PerKO animals, was generated as before and subjected to native anti-FLAG IP. H3.3A complexes were eluted with FLAG peptide and analyzed by mass spectrometry (MS) (Supplementary Fig. 1a). Chromatin from wild-type, non-tagged animals served as negative control and purified fractions were verified by silver staining prior to MS analysis (Supplementary Fig. 1b). Mass spectrometry analysis revealed a significant reduction in specific components of the PBAF complex and the shared ATPase subunit BRG1 in the H3.3A interactome generated from PerKO livers (Supplementary Data 1), suggesting a key role for a H3.3-PBAF complex in establishing the circadian oscillator.

## A circadian PBAF-BMAL1 complex assembles on dynamic H3.3 variant chromatin

We therefore investigated whether the assembly of PBAF-BMAL1 is dynamic on H3.3 variant nucleosomes. To address this, we performed native HA-tag IP on soluble chromatin from FH-H3.3A livers collected at circadian times (every 4 hr over 24 hr in constant darkness, CT0-CT20), using wild-type non-tagged liver chromatin as a negative control. Since the FH-H3.3A animals used in the study are heterozygous, H3.3 nucleosomes could, in principle, retain either a canonical H3 variant or the untagged native H3.3 copy. The absence of canonical histone H3.1/2 in the H3.3 complex across all circadian times indicated that the IP enriches for homotypic H3.3 nucleosomes (Fig. 1b). We found that PBAF remodeling complexes (PBRM1, ARID2 and the shared BRG1 subunit) assemble on H3.3 chromatin independent of circadian time. However, the presence of BMAL1 on H3.3 chromatin was enriched during the daytime, peaking at CT8.

To address whether endogenous PBAF complexes associate with BMAL1, we performed PBRM1 IP from native soluble chromatin from wild-type livers, collected at circadian times CT0-CT20. PBRM1 co-immunoprecipitated with other specific PBAF components, including ARID2, BRD7, and PHF10, as well as the shared ATPase subunit BRG1, across all CTs. However, PBAF-BMAL1 complexes were specifically detected during the active phase at CT4 and CT8 (Fig. 1c), consistent with observations on H3.3A complexes. A control IP experiment using an isotype-matched IgG control showed no specific enrichment of either BMAL1 or PBAF complex proteins at any time point, confirming the specificity of the observed interactions (Fig. 1c).

To determine if H3.3 presence on liver chromatin follows a 24-hour circadian rhythm, we performed native HA-tag Chromatin-Immunoprecipitation Sequencing (ChIP-Seq) on mononucleosome-enriched chromatin from FH-H3.3A livers isolated over circadian time (Supplementary Data 2 and 3). We found that H3.3A deposition exhibits rhythmic presence at CLOCK-BMAL1 target genes and binding sites, with levels steadily increasing from CT20 and peaking at CT4 (Fig. 1d, and Supplementary Fig. 2a). At clock-output genes, well-positioned H3.3A nucleosomes were observed downstream of the transcription start sites (TSS) extending up to nearly 1 kb at all times (Fig. 1d, left panel). In contrast, TSS-upstream nucleosomes were primarily detected during the active phase. Circadian E-box sequences were flanked by cycling H3.3A nucleosomes (Fig. 1d, middle panel), while no such rhythms were detected at short interspersed nuclear element (SINE) repeat regions (Fig. 1d, right panel) or at intergenic regions (Supplementary Fig. 3a). The time of enrichment of H3.3 at clock genes preceded the peak recruitment of CLOCK-BMAL1 and active RNA Polymerase II[2,4,9]. As a control, we performed H3.1/2 ChIP-Seq at CT8 and CT20. Unlike canonical H3 nucleosomes, H3.3A nucleosomes were specifically enriched at TSSs (Fig. 1e) including clock-controlled genes (Supplementary Fig. 3b).

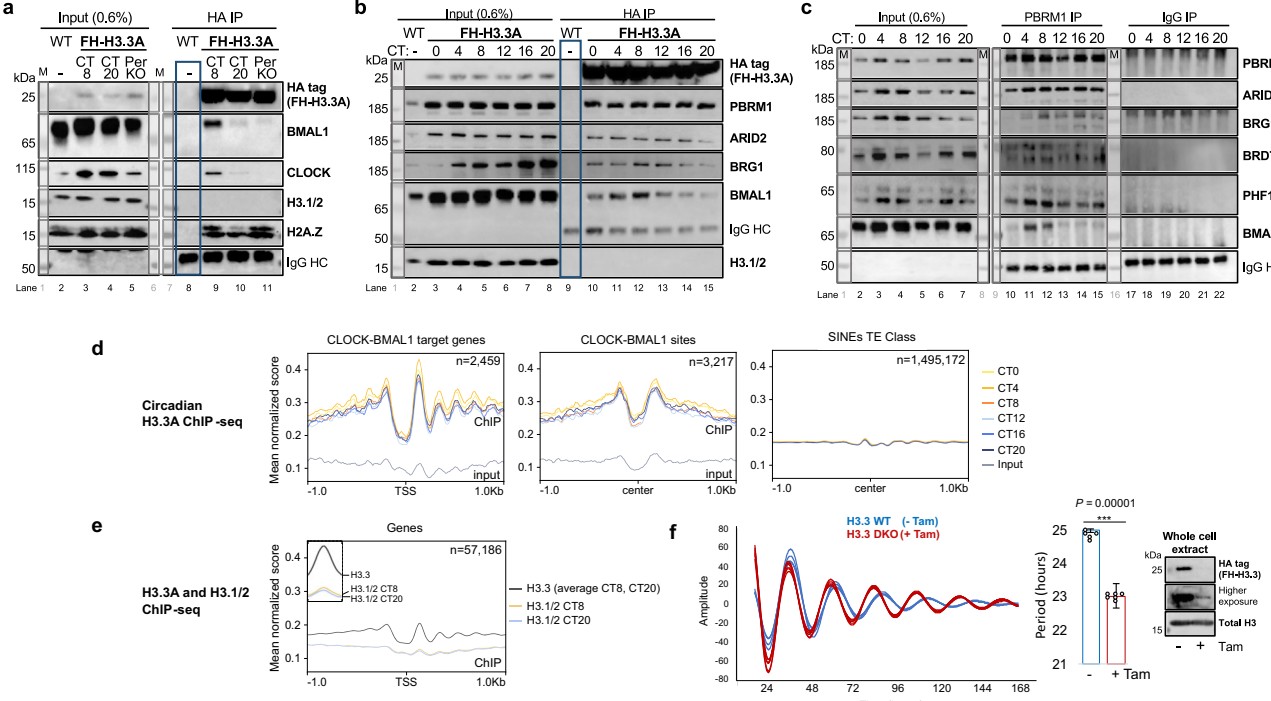

**Fig. 1 | Circadian PBAF-BMAL1 complex assembly peaks on dynamic H3.3 chromatin during the active clock phase. a** Native HA-tag IP on soluble FH-H3.3A chromatin at circadian times (CT) CT8 and CT20 from wild-type and FH-H3.3A;PerKO mouse livers. Wild-type chromatin was used as a negative control (blue box). Nuclear extracts (0.6%) were used as Input (lanes 2-5). Immunoblots of HA (FH-H3.3A), CLOCK, BMAL1, H3.1/2 and H2A.Z. M, protein ladder. Results represent $n = 3$ independent biological replicates, see Source data. **b** Native HA-tag IP on soluble FH-H3.3A chromatin from CT0-20 (lanes 9–15), with wild-type chromatin used as a negative control (blue box). Nuclear extracts (0.6%) were loaded as Input (lanes 2–8). Immunoblots of HA (FH-H3.3A), PBRM1, ARID2, BRG1, BMAL1 and H3.1/2. Results represent $n = 3$ independent biological replicates. **c** Native PBRM1 IP on soluble wild-type liver chromatin over CT (lanes 10-15); IgG antibody was used as a negative control (lanes 17-22). Nuclear extracts (0.6%) were loaded on a separate gel as Input (lanes 2–7). Immunoblots of PBRM1, ARID2, BRD7, PHF10, BRG1 and BMAL1. Results represent $n = 3$ independent biological replicates. **d** Native FH-H3.3A circadian ChIP-seq. Input signal was plotted along with the normalized ChIP signal on the same graph. Y-axis corresponds to the mean of normalized scores per genomic region. X-axis represents the distance from the TSS or the center of the given site/region (±1 kb). Each color represents a given CT, with yellow/orange for the day- and blue colors for the night-time points. Results represent $n = 3$ independent biological replicates, see Supplementary Data 3. **e** Mean H3.3A and H3.1/2 occupancy at day- vs. night-time points at TSS of all genes. ChIP-seq graph axes are as in (d). Results represent $n = 3$ independent biological replicates. **f** Real-time bioluminescence recordings of synchronized wild-type (blue) and H3.3 double knock-out (red) mouse embryonic fibroblasts expressing Bmal1:Luciferase reporter. Detrended analysis over 7 d of measurements is shown. Results represent $n = 5$ (wild-type) and $n = 6$ (knock-out) replicates; the bar graph shows the calculated period length represented as mean ± s.e.m. with individual points indicated for each replicate. P-value between wild-type and H3.3DKO was calculated with paired, one-tailed t-test; ***$P = 1.12542E\text{-}05$. Knock-out efficiency was confirmed by western-blotting. Source Data are provided as a Source Data file.

H3.3 is encoded primarily by 2 genes, H3f3a and H3f3b, in both human and mouse genomes. In mice, additional subsidiary H3.3 variants exhibiting tissue-specific expression are also present[29]. As both *H3f3a* and *H3f3b* encode identical H3.3 proteins, we generated H3.3A and H3.3B knockout mouse embryonic fibroblasts (MEFs) by crossing homozygous floxed strains. In order to determine if H3.3 is important for clock function, double-floxed MEFs were stably transduced with a Bmal1:Luciferase reporter and a Tamoxifen-inducible Cre recombinase. The addition of Tamoxifen to growing fibroblast cultures resulted in a significant reduction of H3.3 protein levels within 4 days (Fig. 1f, Source Data file). We then recorded bioluminescence from the confluent cultures of fibroblasts starting on the fourth day of tamoxifen treatment, for 7 consecutive days. H3.3 depletion caused a significant shortening of the circadian period by nearly 2 hours compared to H3.3 wild-type (floxed) controls (Fig. 1f), indicating a critical role for the H3.3 variant in establishing core oscillator function.

## H3.3-labeled nucleosomes are enriched at CLOCK-BMAL1 target gene TSS in PerKO livers

To investigate whether the absence of circadian negative feedback alters H3.3 chromatin landscapes in the liver, we performed native HA-tag Chromatin Immunoprecipitation Sequencing (ChIP-Seq) on FH-H3.3A;PerKO livers, alongside a control ChIP-Seq for canonical H3.1/2

as previously described. Compared to H3.1/2 nucleosomes, we found that H3.3A deposition was enriched in PerKO livers (Fig. 2a, and left panel, Supplementary Fig. 2a, Fig. 2d upper panel). Notably, H3.3A enrichment in PerKO livers was higher than in wild-type livers at both CT8 and CT20 (Fig. 2a, middle panel), as well as across all circadian times and input samples (Supplementary Fig. 4a, and upper panel). In contrast, SINE repeats and intergenic regions showed no such enrichment (Supplementary Fig. 4a, and lower panel, and Supplementary Fig. 4b).

H3.3A was highly enriched at CLOCK-BMAL1 target genes (Fig. 2a, and right panel, Fig. 2d). However, it is known that not all CLOCK-BMAL1 target genes are expressed in the same phase[30]. To further examine this, we extracted H3.3A occupancy at transcription start sites generating rhythmic transcripts that were either in-phase or out-of-phase with BMAL1 binding, as well as arrhythmic transcripts[30]. While overall enrichment of H3.3A-labeled nucleosomes was noted in all three transcript subclasses in PerKO chromatin, differences in nucleosome positioning were apparent between the classes (Fig. 2b). Well-positioned H3.3A nucleosomes were enriched downstream of the TSS loci that generate transcripts in-phase with BMAL1 binding, whereas the two other categories displayed strong TSS-flanking nucleosomes (Fig. 2b, left and middle panels). The H3.3 variant is also known to mark MNase sensitive CTCF sites, particularly in the context of dual-variant nucleosomes containing H2A.Z at insulators

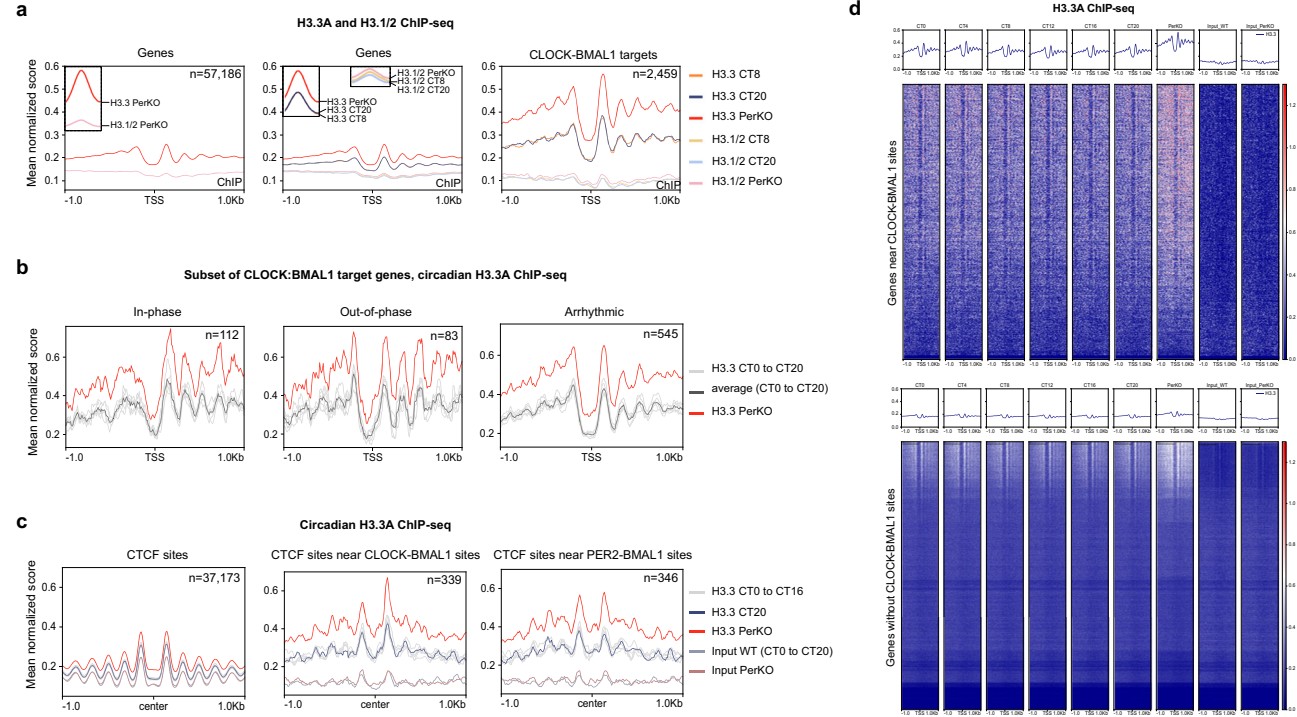

**Fig. 2 | Circadian disruption results in an increase of H3.3-labeled nucleosomes on chromatin. a** Native FH-H3.3A and H3.1/2 ChIP-seq in FH-H3.3A wild-type and PerKO livers at TSS of all genes and CLOCK-BMAL1 target genes, plotted on the same graph. *Y* axis corresponds to the mean of normalized scores per genomic region. *X* axis represents the distance from the TSS (±1 kb). Results were obtained from *n* = 3 independent biological replicates, see Supplementary Data 3. **b** Mean H3.3A occupancy at TSS of CLOCK-BMAL1 targeted rhythmic transcripts in-phase, out-of-phase and arrhythmic transcripts in wild-type and PerKO. ChIP-seq graph axes were represented as indicated in (**a**). Results were obtained from *n* = 3

independent biological replicates. **c** Mean H3.3A occupancy at CTCF sites alone or CTCF sites overlapping with CLOCK-BMAL1 and PER2:BMAL1 binding sites. Graph axes were represented as indicated previously. Results were obtained from *n* = 3 independent biological replicates. **d** H3.3A occupancy exclusively at genes near CLOCK-BMAL1 binding sites (upper panel) and H3.3A occupancy at genes with no CLOCK-BMAL1 sites near (lower panel). *Y* axis corresponds to the mean of normalized scores per genomic regions. *X* axis represents the gene distance in bp, from the TSS (±1 kb). Results were obtained from *n* = 3 independent biological replicates.

and around TSSs[31–33]. In PerKO livers, H3.3A was highly enriched at CTCF sites associated with target genes that contained either CLOCK-BMAL1 binding sites or were co-enriched for PER2 and BMAL1 relative to the genome-wide background (Fig. 2c).

### Loss of ARID2 leads to PBAF complex disassembly in the absence of PER

The assembly of PBAF and cBAF chromatin remodeler complexes occurs in a stepwise manner, with incorporation of ARID subunits serving as a critical determinant of complex identity: ARID2 is essential for PBAF assembly, whereas ARID1A/B directs cBAF formation[34]. Since we observed a significant decrease in key PBAF components in FH-H3.3A complexes in the PerKO livers (Supplementary Data 1), we hypothesized that PBAF assembly on H3.3 chromatin might be altered in these mice. To investigate this, we performed native HA-tag IP experiments to isolate H3.3A-associated complexes from FH-H3.3A livers at CT8 and CT20, as well as from FH-H3.3A;PerKO and non-tagged wild-type liver controls. Consistent with previous results (Fig. 1b), H3.3A interaction with specific PBAF components and BRG1 was slightly higher at CT8 compared to CT20 with BMAL1 present during the daytime timepoint (Fig. 3a, Source Data file). However, these interactions were markedly reduced in PerKO livers. Notably, this loss of interaction was accompanied by a near-total depletion of ARID2 and a substantial reduction in BRG1 in PerKO nuclear extracts, as seen in the input lanes (Fig. 3a, lane 5).

Despite the reduction in ARID2 and BRG1, total PBRM1 protein levels remained mostly unaffected in PerKO tissues (Fig. 3a, b). To confirm the depletion of PBAF complexes in PerKO livers, we

immunoprecipitated endogenous PBRM1-containing complexes from FH-H3.3A and FH-H3.3A;PerKO livers collected at CT8 and CT20, using an isotype-matched IgG control. In wild-type livers, PBRM1-associated PBAF complexes were enriched for H3.3A at CT8 (Fig. 3a, and Supplementary Fig. 5a), while H2A.Z was detected at both CT8 and CT20 (Supplementary Fig. 5a). Indeed, BMAL1 was found only in the daytime PBAF complexes (Fig. 3b). In PerKO livers, where PBAF complexes were absent, BMAL1 was no longer detectable in PBRM1 complexes (Fig. 3b). Moreover, PBRM1 no longer interacted with H3.3A or H2A.Z in PerKO livers (Supplementary Fig. 5a). These findings suggest that PBAF and BMAL1 complexes predominantly associate with daytime chromatin marked by H3.3 to initiate circadian transcription.

The observed reduction in several PBAF and cBAF subunits in PerKO livers appears to be independent of transcriptional regulation. No significant differences in mRNA levels were observed between wild-type and PerKO livers for the key components *Pbrm1*, *Arid2*, *Brd7*, *Phf10*, and the *Brg1* ATPase subunit (Fig. 3c). As a control, *Per1* mRNA levels were depleted as expected, consistent with the disruption of circadian feedback in PerKO tissues. These findings suggest that the coordinated reduction of PBAF and cBAF subunits in PerKO livers likely arises from post-transcriptional mechanisms. This is consistent with previous reports implicating similar pathways in chromatin remodeler subunit dynamics[34,35].

### A remodeler reorganization to cBAF/BRM complexes marks the clock-disrupted state in PerKO livers

In melanoma cell lines, ARID2 depletion leads to PBAF complex disassembly, redistribution of subunits, and a remodeler switch from

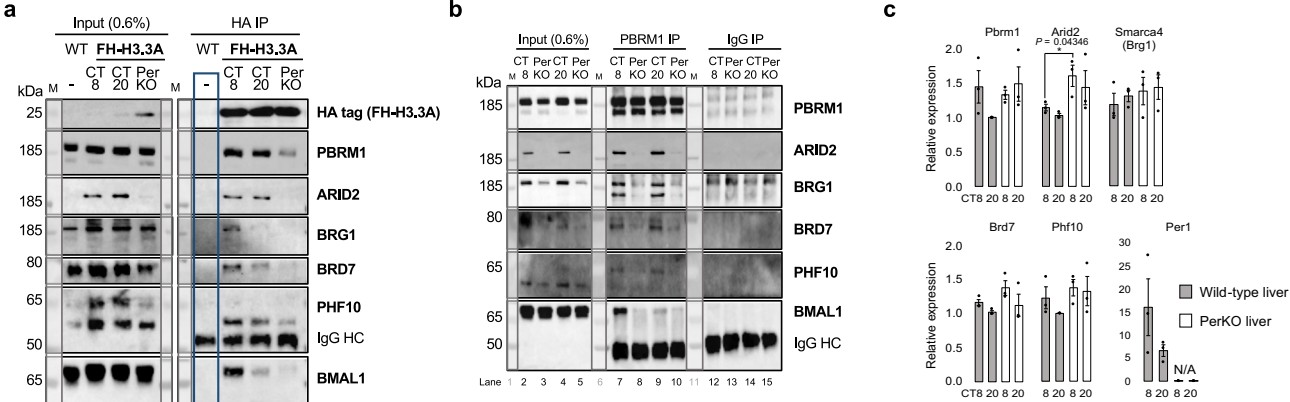

**Fig. 3 | Loss of ARID2 underlies PBAF complex disassembly in the absence of PER. a** Native HA-tag IP at day vs. night time-points in FH-H3.3A wild-type and PerKO livers (lanes 8–11), with wild-type chromatin used as a negative control (blue box). 0.6% of the nuclear extracts were loaded as Input (lanes 2–5). Immunoblots of HA epitope (FH-H3.3A), PBRM1, ARID2, BRD7, PHF10, BRG1 and BMAL1. M, protein ladder. Results are representative of *n* = 3 independent biological replicates, see Source data. **b** Native PBRM1 IP at day vs. night time-points in FH-H3.3A wild-type and PerKO livers (lanes 7–10); IgG antibody was used as a negative control (lanes 12–15). 0.6% of the nuclear extracts were loaded as Input (lanes 2–5). Immunoblots of PBRM1, ARID2, BRD7, PHF10, BRG1 and BMAL1. Results are representative of *n* = 4 independent biological replicates. **c** mRNA relative expression levels of specific PBAF components and shared ATPase subunit *Brg1* normalized to *Rps9*; *Per1* was used as control. Results are represented as mean ± s.e.m. with individual points indicated for each replicate. *P*-values were calculated for all PBAF components between wild-type CT8 and PerKO CT8, and between wild-type CT20 and PerKO CT20 with paired, one-tailed Welch's (unequal variance) t-test; *P* values were not significant, except for *Arid2* CT8 time point; \**P* = 0.04346. Results are representative of *n* = 3 independent biological replicates. Source Data are provided as a Source Data file.

PBAF to cBAF complexes. This switch is associated with increased chromatin accessibility and enhanced transcription factor binding[36]. Given the reduction of ARID2 in PerKO soluble chromatin, we hypothesized that a similar 'switch' to BRM-associated remodeler complexes might occur to regulate BMAL1 function in the absence of circadian negative feedback.

To investigate this, we performed native BRG1 and BRM IPs on soluble chromatin from FH-H3.3A wildtype and FH-H3.3A;PerKO liver chromatin as before and probed the eluates for the presence of BMAL1 and ARID1B (Fig. 4a, b). In PerKO livers, BRG1 and ARID1B levels were decreased in input fractions, leading to reduced assembly of cBAF/BRG1 complexes (Fig. 4a). Consistent with previous findings, BRG1 complexes co-immunoprecipitated BMAL1 during the active clock phase at CT8 with a reduced interaction at CT20[23]. However, BMAL1 was nearly absent in BRG1 complexes from PerKO livers (Fig. 4a).

Interestingly, despite the clear reduction of ARID1B subunit in total protein levels in PerKO input fraction, cBAF/BRM complex assembly remained unaffected in these animals (Fig. 4b). BMAL1 co-immunoprecipitated with BRM at CT8 and CT20, and surprisingly, even in PerKO chromatin (Fig. 4b). BRG1 ChIP-qPCR experiments showed differential enrichment between wildtype and PerKO livers, while no significant difference was seen in BRM ChIP-qPCR experiments at E-boxes of clock genes such as *Per1, Per2, Cry2* as well as immediate target genes *Dbp, Rev-erbα* and *Rorc* (Fig. 4c, d). These findings suggest that ARID2 and BRG1 depletion in PerKO livers disrupts PBAF and cBAF/BRG1 complex assembly, favoring the reorganization of remodelers to cBAF/BRM complexes. This transition could likely target BMAL1 at E-boxes in the absence of negative feedback.

## CLOCK-BMAL1 is recruited to fragile nucleosomes
Among the mSWI/SNF remodeler family, active PBAF complexes exhibit a strong preference for histone H3 post-translational modifications and dual-variant nucleosomes containing H2A.Z and H3.3[37]. Additionally, acetylation marks such as H3K115ac and H3K122ac, located near the lateral surface of the nucleosome, are associated with active chromatin states[21,22]. CLOCK-BMAL1 is known to function as a pioneer transcription factor by binding directly to nucleosomes,

interacting with the acidic patch, and remodeling chromatin[2,3]. We hypothesized that CLOCK-BMAL1 recognizes fragile nucleosomes marked by H3K115ac and H3K122ac and sought to determine whether this interaction is altered in PerKO livers.

To address this, we probed HA-tag immunoprecipitates from FH-H3.3A and FH-H3.3A;PerKO chromatin for these specific acetyl marks, as well as H3 lysine 4 methylation (H3K4me3) that marks active promoters (Fig. 5a). Homotypic H3.3 nucleosomes were marked with active histone modifications, including H3K4me3 and fragile H3K115ac/H3K122ac marks, during the active clock phase at CT8. A near complete loss of these marks was observed 12 h later, at CT20 during the repressive phase (Fig. 5a).

We noted that chromatin from PerKO livers was enriched for fragile nucleosome marks, including H3K4me3, H3K115ac, and H3K122ac, reflecting a persistent daytime-like 'active' chromatin state (Fig. 5a). Despite this enrichment, CLOCK and BMAL1 only co-immunoprecipitated with fragile H3K115ac/H3K122ac-marked homotypic H3.3 nucleosomes during the active clock phase in wild-type livers. Moreover, CLOCK and BMAL1 failed to bind fragile H3.3 nucleosomes in PerKO (Fig. 5a), despite the 'daytime-like' dual-variant chromatin state.

FH-H3.3 nucleosomes do not co-immunoprecipitate canonical H3.1/2 histones, suggesting that CLOCK-BMAL1 binding preferably favors the H3.3 variant nucleosomes and their subsequent remodeling by PBAF/cBAF complexes in wildtype livers. Interestingly, the observed increase in H3.3 deposition in PerKO chromatin (Fig. 2a, b) was not reflected in higher levels of H3.3A mRNA expression (Supplementary Fig. 6a). Total H3.3A protein expression remained unchanged across most replicates (N > 6, Supplementary Fig. 6a). We therefore wondered if changes in H3.3-specific chaperones could explain the higher loading of H3.3 in PerKO chromatin. However, nuclear protein levels of the primary H3.3 chaperones, HIRA and DAXX, were not consistently altered across multiple samples (Supplementary Fig. 6b).

Therefore, we wondered if H3K115ac marked nucleosomes retained CLOCK and BMAL1 in PerKO chromatin. To address this question, we purified endogenous H3K115ac nucleosome complexes from FH-H3.3A and FH-H3.3A;PerKO livers (Fig. 5b) using an antibody that fails to distinguish between canonical H3.1/2 and the H3.3 variant.

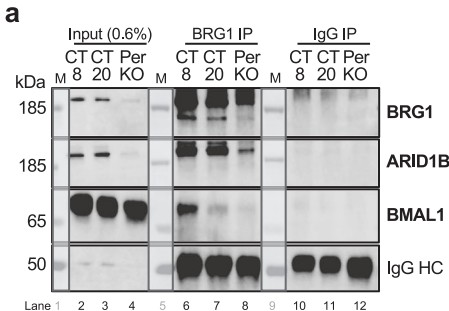

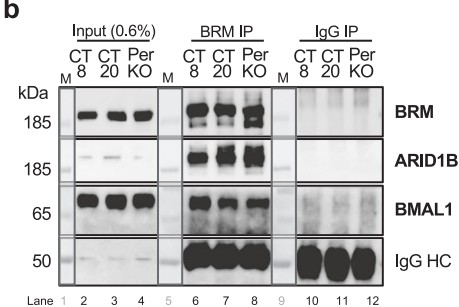

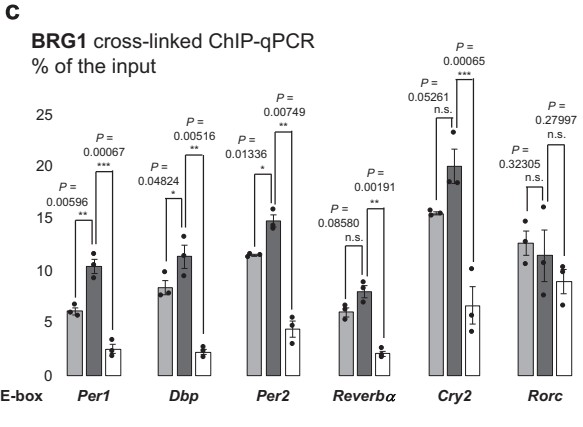

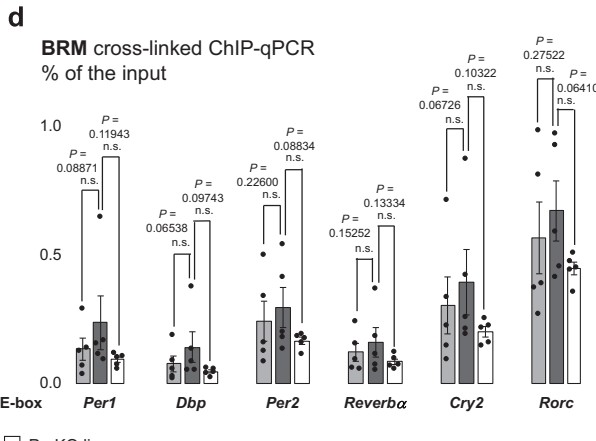

**Fig. 4 | Clock-disrupted state is marked by PBAF-cBAF/BRG1 to cBAF/BRM remodeler reorganization. a** Native BRG1 IP at day vs. night time-points in FH-H3.3A wild-type and PerKO livers (lanes 6–8); IgG antibody was used as a negative control (lanes 10-12). 0.6% of the nuclear extracts were loaded as Input (lanes 2–4). Immunoblots of BRG1, ARID1B and BMAL1. M, protein ladder. Results are representative of $n = 3$ independent biological replicates, see Source data. **b** Native BRM IP at day vs. night time-points in FH-H3.3A wild-type and PerKO livers (lanes 6-8); IgG antibody was used as a negative control (lanes 10-12). 0.6% of the nuclear

extracts were loaded as Input (lanes 2–4). Immunoblots of BRM, ARID1B and BMAL1. M, protein ladder. Results are representative of $n = 3$ independent biological replicates. **c, d** BRG1 and BRM cross-linked ChIP-qPCR. Enrichment at *Per1*, *Dbp*, *Per2*, *Rev-erbα*, *Cry2* and *Rorc* E-boxes was assessed, calculated as percentage of the input. Results are represented as the average ± s.e.m. of $n = 3$ independent biological replicates. *P*-values were calculated either with paired, one-tailed t-test or with paired, one-tailed Welch's (unequal variance) t-test and are indicated in the figures. Source Data are provided as a Source Data file.

The H3K115ac pull-down revealed fragile nucleosomes that harbor either the H3.3 variant and/or H3.1/2 canonical histones (Fig. 5b). CLOCK and BMAL1 proteins consistently associate with H3K115ac-fragile nucleosomes at subjective day and night time-points in the wild-type, as well in PerKO chromatin. In addition, H3K115ac co-immunoprecipitated with H3.3A, H2A.Z, and H3K122ac marks in all conditions. These findings indicate that circadian transcription factor complexes interact with canonical and variant H3 nucleosomes that are inherently fragile and mobile.

Finally, we analyzed H3K115ac complexes for components of the SWI/SNF remodeler subunits. Fragile H3K115ac nucleosomes retained PBAF (PBRM1, ARID2) and shared BRG1/BRM ATPase sub-units at both CT8 and CT20 (Fig. 5b). However, ARID2-mediated PBAF assembly and cBAF/BRG1 complexes were disrupted on H3K115ac nucleosomes in PerKO chromatin, consistent with the general reduction in ARID2 and BRG1 proteins observed earlier (Fig. 5b, lane 9 and Fig. 3a, b). Remarkably, cBAF/BRM complex assembly on H3K115ac nucleosomes remained unaffected in PerKO chromatin, suggesting a preferential interaction of BRM remodelers with fragile canonical or variant H3 nucleosomes.

### Fragile H3.3 chromatin in PerKO livers enhances E-box accessibility

Does an increased H3.3 deposition in PerKO chromatin and the inter-action with BRM remodelers impact BMAL1 binding in the absence of negative feedback? To test this, we performed native BMAL1 ChIP-seq

on soluble mononucleosome-enriched chromatin from FH-H3.3A (CT8 and CT20) and FH-H3.3A;PerKO livers. Unlike crosslinked BMAL1 ChIP-seq[4], we observed only a weak enrichment of BMAL1 on chromatin. Nevertheless, BMAL1 was enriched at H3.3 nucleosome positions that flank CLOCK-BMAL1 sites (Fig. 5c), with a greater signal at CT8 centered on binding site as well as on flanking nucleosomes. Strikingly, PerKO chromatin showed significantly enhanced BMAL1 enrichment centered at CLOCK-BMAL1 sites (Fig. 5c). These findings suggest that the fragile H3.3 chromatin landscape and altered remodeling dynamics in PerKO livers render E-box motifs more accessible. While H3.3 nucleosomes at CT8 could differ from those in PerKO livers in their PTM landscapes or the presence of other histone variants, the reduced association of BMAL1 with H3.3 complexes in PerKO chromatin (Fig. 3a) likely reflects BMAL1 directly binding E-boxes or an increased turnover at these sites.

## Discussion

Our findings reveal that H3.3-labeled fragile nucleosomes play a pivotal role in establishing chromatin states that facilitate CLOCK-BMAL1 heterodimer binding and activation of circadian transcription. These nucleosomes, remodeled by the PBAF/cBAF-BRG1 complex, contribute to rhythmic chromatin accessibility (Fig. 5d). H3.3 deposition peaks between CT4 and CT8, coinciding with the active transcription phase, and is marked by PTMs at the dyad, such as H3K115ac and H3K122ac, which enhance nucleosome mobility. Depletion of H3.3 in fibroblasts results in a decrease in clock gene expression (Supplementary Fig. 7a)

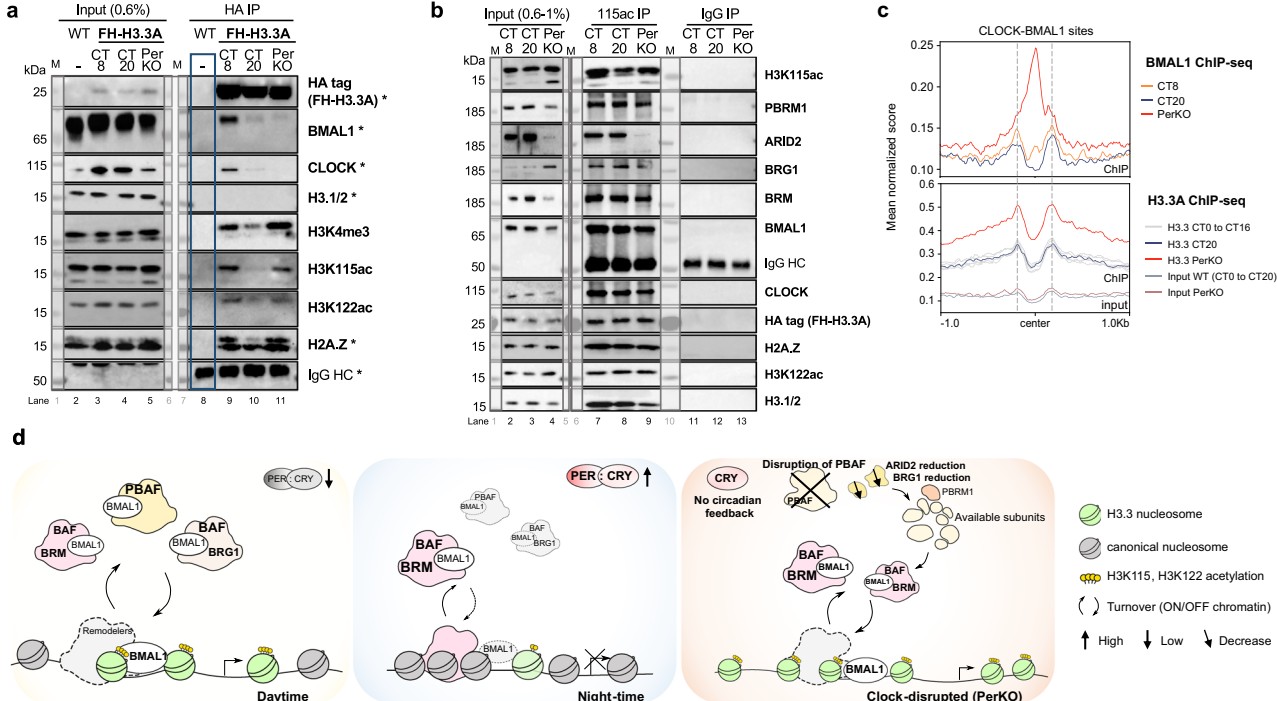

**Fig. 5 | H3K115ac marked fragile chromatin enhances E-box accessibility and CLOCK-BMAL1 binding in the absence of PER. a** Native HA-tag IP at day vs. night time-points in FH-H3.3A wild-type and PerKO livers (lanes 8–11), with wild-type chromatin used as a negative control (blue box). 0.6% of the nuclear extracts were loaded as Input (lanes 2–5). Immunoblots of HA epitope (FH-H3.3A), BMAL1, CLOCK, H3.1/2, H2A.Z as shown in Fig. 1a *marked with \**. Samples were analyzed for H3K4me3, H3K115ac, and H3K122ac. M, protein ladder. Results are representative of *n* = 3 independent biological replicates, see Source data. **b** Native H3K115ac IP at day vs. night time-points in FH-H3.3A wild-type and PerKO livers (lanes 7–9); IgG antibody was used as a negative control (lanes 11-13). 0.6-1% of the nuclear extracts were loaded as Input (lanes 2–4). Immunoblots of H3K115ac, PBAF/cBAF components, BMAL1, CLOCK, HA epitope (FH-H3.3A), H2A.Z, H3K122ac and H3.1/2. M, protein ladder. Results are representative of *n* = 3 independent biological replicates. **c** Mean BMAL1 and H3.3A occupancy at circadian E-boxes at CT8 and CT20 on FH-H3.3A wild-type and PerKO chromatin. *Y* axis corresponds to the mean of

normalized scores per genomic regions. *X*-axis represents the distance from the center of CLOCK-BMAL1 binding site ( ± 1 kb). Each color represents a given CT, with orange for the day-, blue for the night-time points, and red for the PerKO. Results were obtained from *n* = 3 independent biological replicates, see Supplementary Data 3. Source Data are provided as a Source Data file. **d** Proposed model illustrating changes in BMAL1 targeting H3.3 variant chromatin in wildtype animals and in the absence of circadian negative feedback. Under physiological conditions, fragile H3.3 nucleosomes are enriched at TSS and CLOCK-BMAL1 binding sites during the active clock phase. These nucleosomes serve as regulatory hubs for assembly of BMAL1-SWI/SNF chromatin remodeling complexes (PBAF, BRG1/cBAF and BRM/cBAF). BMAL1-BRM/cBAF complexes persist during the repressive clock phase. Upon circadian clock disruption, an enrichment of fragile H3.3 nucleosomes is coincident with the loss of PBAF and BRG1/cBAF complexes. A BRM/cBAF remodeler targets BMAL1 TF activity at more accessible genomic binding sites.

supporting a role in activating transcription. H3K4me3 is also enriched at CT8 as expected from the daytime recruitment of the MLL complex[38]. Consistent with previous reports, BMAL1 interacts rhythmically with cBAF/BRG1 on H3.3 chromatin, peaking at CT8[23]. However, at CT20, BMAL1 remains associated with cBAF/BRM remodelers and fragile H3 nucleosomes. BRM catalytic activity has been shown to increase nucleosome density at clock gene promoters in *Drosophila* to limit transcriptional activity during the active phase while a non-catalytic activity could limit core-clock transcription factor binding to target promoters during the repressive phase[8]. Thus, BRM-BMAL1 complexes might be non-functional at CT20 and promote off-chromatin TF states. Alternatively, the complex may play a role in priming circadian loci for H3.3 incorporation and activation, possibly via the CHD4 subunit of the NuRD complex. Notably, repressor components of the clock (PERs and CRYs) were not detected in the H3.3 - PBAF/cBAF-BRG1 complex.

In PerKO liver tissue, the loss of PBAF-cBAF/BRG1 complex assembly leads to a cBAF/BRM remodeling complex reorganization and strong BMAL1 targeting to E-boxes. These clock binding sites are presumably being rendered more accessible within linker DNA by the increased incorporation of fragile H3.3 nucleosomes. Intriguingly, while homotypic H3.3 nucleosomes were not enriched for BMAL1 in PerKO chromatin, CLOCK-BMAL1 continued to be recruited to H3K115ac marked H3 nucleosomes. In a similar vein, the

persistent interaction of BMAL1 with cBAF/BRM remodeler could also be a chromatin mechanism to temper BMAL1 activity akin to what has been proposed in the *Drosophila* clock at specific clock genes[8]. Indeed, while BMAL1 protein level is reduced generally in PerKO tissues, presumably due to protein turnover linked to activity, it is appreciably enriched in PerKO chromatin compared to CT20 (Supplementary Fig. 8a). Proteasome inhibitor treatment of PerKO fibroblasts further confirmed this hypothesis, as BMAL1 signals increased on chromatin following MG132 treatment (Supplementary Fig. 8b).

Three related SWI/SNF complexes exist in mammalian cells, designated canonical cBAF (BRG1- or BRM-associated factors), non-canonical ncBAF and PBAF (Polybromo-associated BAF). Recently, Gourisankar and colleagues discovered that BAF/PBAF chromatin remodelers can be found in three different states prior activation - as fully assembled functional complexes, partially assembled in their poised state, or as unassembled subunits - in neurons and MEFs. Upon stimulation, rapid large-scale reorganization of these complexes helps assemble the majority of functional remodelers[39]. The PBAF complex is distinguished by the presence of specific subunits PBRM1, ARID2, BRD7 and PHF10, in addition to other shared subunits. The core factor PBRM1 is partially enriched at CLOCK-BMAL1 sites[40]. Moreover, in the facilitated-repression model proposed by Zhu and colleagues, PBAF has been shown to help the loading of REV-ERB alpha on the hepatic

genome at the end of the activation phase. Thus, the loss of PBAF in PerKO tissues likely disrupts auxiliary transcriptional feedback loops and REV-ERB targeting on the genome and contributes to metabolic and physiological alterations in the liver. We also observed a marked reduction in ARID1B-containing cBAF complexes in PerKO livers, mirroring findings in other systems where BAF/ARID1A depletion increased H3.3 occupancy at the TSS[41]. A similar mechanism might also explain pervasive H3.3 enrichment in PerKO chromatin, likely driven by altered chromatin remodeling.

Unlike replication dependent H3.1/2 histones, H3.3 is expressed throughout the cell cycle and is progressively enriched in chromatin from the liver, brain and other tissues as mice age[42,43]. Remarkably, we find a global increase in H3.3 deposition in PerKO liver chromatin, suggestive of a higher rate of aging in the liver. The enrichment is coincident with global changes in H3.3 post-translational modifications such as methylations at K36me2 and K4me1 positions[43]. In particular, the hypothalamic expression of a H3.3Q5A mutant, which alters methylation rates at the adjacent K4 position disrupted behavioral rhythms and diurnal gene expression suggesting a role for H3.3 function in rhythmic transcription[44].

siRNA-mediated knockdowns of specific PBAF/cBAF components and the analysis of published datasets[45] indicated that SWI/SNF subunit depletion has limited direct effects on clock gene expression and cellular rhythms (Supplementary Fig. 9a–h, S10a). Depletion of *Arid1b*, *Brg1* and *Brd7* in a U2OS-Bmal1:Luciferase cell line reduced *Per1* and *Per2* expression, while *Pbrm1* and *Brm* depletion led to increase in *Bmal1* and *Cry1* expression (Supplementary Fig. 9a–h). *Brg1* depletion was shown to increase the circadian period in U2OS cells[23] but these experiments did not reflect data obtained from larger siRNA screens where no effect was seen[45]. ChIP experiments between day and night for BRG1 did not correlate with the observed remodeler complex assembly data. Indeed, the results of our BRG1 ChIP-qPCR experiments are different from what has been published[23], which support our findings that BRG1 complexes assemble preferentially at CT8. The differences could result from antibody specificity, or from the shorter crosslinking times in our ChIP experiments. The discrepancy probably also arises from purification of soluble remodeler complexes that exist both on- and off-chromatin while ChIP experiments would enrich only for on-chromatin signals for the remodeler.

To a large extent however, the clock remains robust to loss of other remodeler subunits (Supplementary Fig. 10a). This robustness likely stems from functional redundancy among SWI/SNF complexes, allowing re-purposing of remaining subunits to maintain functionality[34,36,39]. Similarly, genetic redundancy within histone variants provides a buffering mechanism. For example, loss of individual H2A.Z variant paralogs in mouse fibroblast cultures[15] had minimal impact on cellular rhythms, as compensatory upregulation of paralogs or subvariants preserves chromatin integrity and clock function. This epigenetic buffering, akin to mechanisms observed in core-clock genes[46], underscores the evolutionary robustness of the circadian machinery.

Last but not least, PER proteins are thought to act as tumor suppressors and *Per1$^{-/-}$;Per2$^{-/-}$* double knockout animals exhibit a three-fold increase in hepatocellular carcinomas under chronic jetlag[47]. Notably, BAF/PBAF remodeler components are mutated in ~20% of cancers, with recurrent alterations in ARID2, PBRM1, and BRG1 subunits[35]. Furthermore, H3.3 driver mutations have been implicated in glioblastomas, bone cancers, and other malignancies[48,49]. Our data suggest that oscillator dysfunction could contribute to chromatin reorganization that is characteristic of the cancer epigenome. Circadian disruption, which is pervasive in modern society, may not only predispose individuals to such epigenomic reorganization but also result in accelerated progression of the disease.

## Methods

### Mouse models and tissue collections

Genetically modified heterozygous FLAG-FLAG-HA-tag (FH) - H3.3A$^{+/-}$, FH-H3.3A$^{+/-}$ *Per1$^{-/-}$*, *Per2$^{-/-}$* knock-out mouse lines and wild-type (WT) C57Bl/6J male mice were used in this study. The FH-H3.3A$^{+/-}$ mouse line has been generated as described previously[28]. FH-H3.3A$^{+/-}$ littermates were obtained by crossing FH-H3.3A$^{+/-}$ with WT mice. Male FH-H3.3A$^{+/-}$ heterozygous mice were kept for the study. The FH-H3.3A$^{+/-}$ *Per1$^{-/-}$*, *Per2$^{-/-}$* knock-out line was generated by crossing FH-H3.3A$^{+/-}$ with *Per1$^{-/-}$*, *Per2$^{-/-}$* knock-out mice. Animals were housed under a temperature-controlled, 12 h:12 h light-dark cycles, and were fed *ad libitum*. All mice used in this study were from 10- to 17-weeks old male littermates. Aged-matched littermates were randomly assigned to experimental groups. Prior tissue collection, mice were entrained to a 12 h:12 h light-dark cycle in temperature and humidity-controlled ventilated cabinets (~22 °C, 30% RH) for 2 weeks and then kept in constant darkness. Mice were euthanized at CT0, 4, 8, 12, 16 and 20 under the red light, and tissues dissected under the room light. Animal procedures were carried out according to the French guidelines for care and use of experimental animals and approved by the ethical committee of the ENS Lyon and the French ministry of science under APAFIS no. 5212-2016032914413553.

Liver tissues from FH-H3.3A$^{+/-}$ and WT animals were collected at 1 day in constant darkness and from FH-H3.3A$^{+/-}$ *Per1$^{-/-}$*, *Per2$^{-/-}$* KO animals at 1 or 4 days in constant darkness. Livers were mainly processed fresh, followed by nuclei purification immediately after tissue isolation, or fresh-frozen and stored at −80 °C until use. Nuclei were purified as described in ref. 50 and stored in Nuclear Lysis Buffer at 1X final with 10% glycerol (NLB 10X stock: 10 mM HEPES-KOH pH 7.6, 100 mM KCl, 0.1 mM EDTA-NaOH pH 8.0) at −80 °C until use.

### Native purification of H3.3A protein complexes for mass spectrometry analysis

Nuclei isolated from FH-H3.3A$^{+/-}$ livers collected at CT8 and CT20 and FH-H3.3A$^{+/-}$ *Per1$^{-/-}$*, *Per2$^{-/-}$* KO livers collected at CT20 were used for the native purification of H3.3A protein complexes. Nuclei from WT livers collected at random time points were used as negative control for the experiment. Total nuclear protein quantity was estimated by lysing 10ul of purified nuclei with RIPA lysis buffer (Invitrogen) for 20 min on ice and sonicated for 6 cycles on Diagenode bioruptor (30 s on/ 30 s off) at 4 °C, followed by centrifugation at 14,000 rpm (18,407 g) for 10 min. Released proteins in the supernatant were quantified using DC protein assay (Bio-Rad, 5000116).

MNase-digested nuclear extracts were prepared from one full liver (~1.4 mg of total protein) for each condition according to Khan et al.[51] with modifications, as follows. Nuclei were washed twice in Buffer A (10 mM HEPES-KOH pH 8.0, 10 mM KCl, 1.5 mM MgCl$_2$, 340 mM Sucrose, 10% Glycerol, 1 mM DTT) containing protease inhibitors and centrifuged at 4500 rpm (1902g) for 4min30. Nuclear pellets were resuspended in Cutting Buffer (15 mM NaCl, 60 mM KCl, 10 mM Tris-HCl pH 7.5, 2 mM CaCl$_2$) and freshly diluted MNase (Nuclease S7, Roche 10107921001, batch number 53964400; resuspended and stored in 10 mM Tris-HCl pH 7.5, 50 mM NaCl, 1 mM EDTA-NaOH pH 8.0, 50% glycerol) in Cutting buffer was added at 10U per sample. To digest chromatin to mononucleosomes, samples were vortexed at low speed and incubated in water-bath at 37 °C for 7 min, vortexed every 1min30. After digestion, samples were kept on ice and MNase activity was stopped by adding EGTA-NaOH pH 8.0 at 20 mM final. Samples were vortexed and centrifuged at 1300 g for 5 min. The totality of the supernatant was kept in separate tubes on ice as S1 MNase-digested fraction. Remaining pellets were resuspended in TE buffer (10 mM Tris-HCl pH 8.0, 1 mM EDTA-NaOH pH 8.0) containing protease inhibitors and incubated on end-to-end rotator at 4 °C for 30 min. After incubation, samples were centrifuged at 13,000 g for 5 min and the totality of the supernatant was kept in separate tubes on ice as S2 TE-soluble

fraction. Salt concentration was adjusted for the totality of both, S1 and S2 fractions as follows. Equal sample volume of 2XE buffer (30 mM HEPES-KOH pH 7.5, 225 mM NaCl, 3 mM MgCl$_2$, 0.4% Triton X-100, 20% Glycerol) for the S1 fraction, and half of the sample volume of 3XD buffer (60 mM HEPES-KOH pH 7.5, 450 mM NaCl, 4.5 mM MgCl$_2$, 0.6 mM EGTA-NaOH pH 8, 0.6% Triton X-100, 30% Glycerol) for the S2 fraction was added dropwise on vortex. Samples were centrifuged at 13,000 g for 5 min and supernatants from both fractions were pooled together respectively for each condition in separate tubes, and kept on ice as nuclear extracts. All centrifugation steps were performed at 4 °C and all vortex steps at the lowest speed.

Prior purification, 5% of each nuclear extract was saved as input for further analysis. Nuclear extracts were incubated with prewashed Anti-FLAG M2 Affinity Gel agarose beads (Sigma-Aldrich), over-night (18 h) on an end-to-end rotator at 4 °C. After incubation, samples were centrifuged at 1500 rpm (211 g), 2 min and the unbound fraction was isolated. Sequential washing steps were applied to remove non-specific binding, as follows. Beads were washed 4 times with 1XD buffer made from 3XD, followed by 4 washes with 1XD buffer made from 3XD + 0.5% Triton X-100, followed by 2 washes with 1XD buffer without Triton X-100. Each washing step was done on rotation at 4 °C for 5 min, followed by centrifugation at 1500 rpm, 2 min. After the final wash, beads were dried and specifically-bound proteins eluted by a tandem elution with the 3xFLAG peptide (Sigma-Aldrich). Both elution steps were performed with 3xFLAG peptide at 500 ug/ml final concentration, resuspended and diluted in TEGN buffer (20 mM Tris-HCl pH 7.65, 0.1 mM EDTA-NaOH pH 8.0, 10% Glycerol, 150 mM NaCl, 0.01% NP-40), for 3 h at room temperature with tilting mode. Both eluates were pooled respectively for each condition and 40% of eluates kept to further assess the specificity of the pull-down by silver staining (SilverQuest Silver Staining kit, Invitrogen) and western-blotting. Remaining eluates were submitted for Mass Spectrometry analysis.

## LC-MS/MS analysis

Label-free quantitative Mass Spectrometry analysis was performed at Protein Sciences Facility - SFR BioSciences (UAR3444/US8) in Lyon, using Q Exactive HF (Thermo Scientific) Mass Spectrometer coupled with nanoUHPLC U3000 (Thermo Scientific). Briefly, samples were reduced and alkylated with tris-(2-carboxyethyl) phosphine/ iodoacetamide (TCEP/IAA) and fixed on SP3 beads prior digestion. Beads were washed with 80% ethanol and protein digested in ammonium bicarbonate (100 mM), using Lys-C/Trypsin protease (enzyme/protein ratio: 1/100) for 18 h at 37 °C. Supernatants with peptides were recovered after centrifugation at 20,000 g for 1 min and peptide concentration was determined for each sample by quantitative fluorometric peptide assay (Thermo Scientific) to inject 400 ng of peptides in the nanoLC-MS/MS system.

The two biological replicates for each condition were injected in triplicates and loaded on a C18 Acclaim PepMap100 trap-column 300 μm ID x 5 mm, 5 μm, 100 Å, (Thermo Scientific) for 3.0 minutes at 20 μL/min with 2% ACN, 0.05% TFA in H2O and then separated on a C18 Acclaim Pepmap100 nano-column, 50 cm×75 μm i.d, 2 μm, 100 Å (Thermo Scientific) with a 60 minutes linear gradient from 3.2% to 40% buffer B (A: 0.1% FA in H2O, B: 0.1% FA in ACN), from 40% to 90% of B in 2 min, hold for 10 min and returned to the initial conditions in 1 min for 14 min. The total duration was set to 90 minutes with a flow rate of 300 nL/min and the oven temperature was kept constant at 40 °C.

Samples were analyzed with a TOP20 DDA HCD method: MS data were acquired in a data dependent strategy selecting the fragmentation events based on the 20 most abundant precursor ions in the survey scan (350-1600 Th). The resolution of the MS scan was 60,000 at m/z 200 Th and for MS/MS scan the resolution was set to 15,000 at m/z 200 Th. Parameters for acquiring HCD MS/MS spectra were as follows: normalized collision energy = 27 and an isolation window width of 2 m/z. The precursors with unknown charge state, charge state of 1 and 8 or greater than 8 were excluded. Peptides selected for MS/MS acquisition were then placed on an exclusion list for 20 s using the dynamic exclusion mode to limit duplicate spectra.

Sample file was treated with Proteome Discoverer 2.5 software (Thermo Scientific), using SEQUEST HT research engine and Swiss-Prot Mus musculus database (august 2021 release, 17500 entries), supplemented with protein contaminant database. Precursor mass tolerance was set at 10 ppm and fragment mass tolerance was set at 0.02 Da, and up to 2 missed cleavages were allowed. Oxidation (M), phosphorylation (S, T, Y), acetylation (Protein N-terminus) were set as variable modifications and carbamidomethylation (C) as fixed modification. False discovery rate (FDR) of peptide identifications was calculated by the Percolator algorithm method, and a cut-off FDR value of 1 % was used. Protein quantification was done by Label Free Quantification (LFQ) approach, and LFQ abundance values were obtained and normalized to the total peptide amount. Protein quantitation was performed with precursor ions quantifier node in Proteome Discoverer 2.5 software with protein quantitation based on pairwise ratios and hypothesis t-test. Proteins were considered as differentially expressed between the two conditions when FC > 2 and p-value < 0.05. Adjusted p-values were obtained by applying the Benjamini-Hochberg procedure. Generated data files were further analyzed taking into account following parameters in order to determine specific candidates of FH-H3.3A pull-down: 'not found in WT', number of peptides, number of unique peptides, Score Sequest HT: Sequest HT, abundance ratio and adjusted p-value abundance ratio for each peptide of given protein.

## Native immunoprecipitation and western-blotting

Native immunoprecipitation experiments were performed on MNase-digested nuclear extracts from wild-type, FH-H3.3A$^{+/-}$ and FH-H3.3A$^{+/-}$ Per1$^{-/-}$, Per2$^{-/-}$ KO nuclei purified from livers collected either every 4 h over 24 h, or at CT8 and CT20. Nuclei from WT livers collected at random timepoints, or isotype matched Anti-Mouse IgG antibody (Jackson ImmunoResearch, 315-005-003) were used as negative controls for the IP.

Nuclear extracts were prepared as above, with few following modifications. ~530ug of total protein were used for the IP, with all buffer volumes reduced by half and diluted MNase was used at 5U per sample. Prior IP, 5% of each nuclear extract was saved as input, mixed with 1.5X SDS-blue loading buffer containing DTT at 100 mM, denatured at 80 °C for 7 min and stored at -20 °C for further analysis. Nuclear extracts were incubated overnight (18 h) on an end-to-end rotator at 4 °C with 2 - 4.5ug of following antibodies: anti-HA-Tag (C29F4) (Cell Signaling, #3724); anti-PBRM1/BAF180 (E9X2Z) (Cell Signaling, #89123); anti-BRG1/SMARCA4 (Bethyl Laboratories, A300-813A); anti-SMARCA2 antibody [HL1115] (GENETEX, GTX636330) and anti-Acetyl-Histone H3 (Lys115) (PTM Bio, PTM-170). Following overnight incubation with antibodies, equal volume of Protein A and Protein G Dynabeads (Invitrogen, 10002D, 10004D) mix was added per sample and nuclear extracts were incubated on an end-to-end rotator for 3-4 h at 4 °C. Unbound fraction was isolated and beads were washed 2-3 times with 1XD buffer made from 3XD, on rotation at 4 °C for 5 min. After final wash, beads were resuspended in 2.5X SDS-blue loading buffer containing DTT and proteins eluted for 10 min at room temperature, with shaking mode. Protein eluates were transferred to separate tubes, denatured at 80 °C for 7 min and stored at -20 °C.

SDS-PAGE electrophoresis was performed using precast Novex NuPAGE 4–12% Bis-Tris gradient gels and NuPAGE SDS MOPS Running buffer (20X) (Invitrogen) at 1X final. Prestained Protein Ladder Plus (Euromedex, 06P-0211) was used as a reference marker for protein size. For each IP experiment, 0.6–1% of input and one-third of IP eluates were loaded per lane. Transfer was performed on 0.2μm nitrocellulose membranes (Amersham, GE10600004) at constant current (0.4 A) for 2–3 h at 4 °C, using NuPAGE SDS MOPS Running buffer at 1X with 5% ethanol. After transfer, membranes were rinsed with Millipore water and

incubated in 5% non-fat dry milk- PBS 1X blocking solution for 30 min at room temperature. After blocking, membranes were first rinsed with Millipore water, then with PBS 1X- 0.1% Tween 20 wash buffer prior overnight incubation at 4 °C with respective primary antibodies. After incubation with primary antibodies, membranes were washed 2-3 times with PBS 1X- 0.1% Tween 20 wash buffer for 5-10 min and incubated with respective secondary antibodies for 1 h–1 h.30 at room temperature. Peroxidase Goat Anti-Rabbit IgG (H + L) (Jackson ImmunoResearch, 111-035-144) or Goat Anti-Mouse IgG (H + L)- HRP Conjugate (Bio-Rad, #1706516) secondary antibodies were used at 1:10,000 in 5% milk- PBS 1X. Membranes were washed 2-3 times with PBS 1X- 0.1% Tween 20 wash buffer for 5–10 min, and incubated with ECL Prime western blotting reagent (CYTIVA, RPN2232) prior developing. Image acquisitions were performed using ChemiDoc Imaging system (Bio-Rad) and ImageLab software (version 6.0.1).

Following primary antibodies were used for western-blotting in this study, dilutions were prepared in 3% BSA- PBS 1X as indicated: HA-Tag (C29F4) Rabbit mAb (Cell Signaling, #3724), 1:1000; PBRM1/BAF180 (E9X2Z) Rabbit mAb (Cell Signaling, #89123), 1:1000; ARID2 (GT7311) Mouse mAb (Sigma-Aldrich, SAB2702340), 1:500; ARID2 (GT7311) Mouse mAb (GENETEX, GTX632011), 1:500; BRG1/SMARCA4 Rabbit pAb (Bethyl Laboratories, A300-813A), 1:500; BRD7 Rabbit pAb (Proteintech, 51009-2-AP), 1:500; PHF10 Mouse mAb (Proteintech, 66341-1-Ig), 1:500; PHF10 Rabbit pAb (GENETEX, GTX116314), 1:500; BMAL1 Rabbit pAb (Bethyl Laboratories, A302-616A), 1:1000; CLOCK Mouse mAb (MBL, D334-3), 1:500; Histone H3.1/3.2 (1D4F2) Mouse mAb (Active Motif, 61629), 1:500; Tri-Methyl-Histone H3 (Lys4) Rabbit pAb (Cell Signaling, #9727), 1:500; Acetyl-Histone H3 (Lys115) Rabbit pAb (PTM Bio, PTM-170), 1:1000; Histone H3 (acetyl K122) Rabbit pAb (Abcam, ab33309), 1:500; Histone H2A.Z Rabbit pAb (Active Motif, 39113), 1:1000; Homemade H2A.Z Rabbit pAb (kind gift from Stefan Dimitrov), 1:500; ARID1B Rabbit pAb (GENETEX, GTX130708), 1:500; SMARCA2 (HL1115) Rabbit mAb (GENETEX, GTX636330), 1:500; Histone H3 Rabbit pAb (Proteintech, 17168-1-AP), 1:2000 in 1% milk- PBS 1X; Histone H4 Rabbit pAb (Proteintech, 16047-1-AP), 1:2000 in 1% milk- PBS 1X; U2AF65 Rabbit mAb (Abcam, ab197031), 1:500; HIRA (clone WC119) Mouse mAb (Sigma-Aldrich, 04-1488), 1:500; DAXX anti-Mouse (Precision antibody), 1:500.

If the proteins of interest were migrating at the same size, mainly different gels and membranes were used. In some cases, primary anti-ARID2 mouse antibody was used first and after extensive stringent washing for few hours, membranes were probed in primary anti-BRG1 rabbit antibody (before probing in primary rabbit antibody, extensively washed membranes were incubated with the anti-mouse secondary antibody and developed to verify the removal of the signal). Membranes were probed in the following order: PBRM1 and CLOCK (on the same membranes); ARID2, BRG1 on different membranes (and both on the same membranes in Fig. 1a replicate 1 and 2, Fig. 1b replicate 2, Fig. 3a replicate 1 and 3, Fig. 3b replicate 1 and 2; see Source Data file for the replicates); BMAL1, BRD7 on different membranes (and on the same membrane in Fig. 3a replicate 1); PHF10 on different membranes; ARID1B, SMARCA2 (BRM) on different membranes; the histones and histone marks were probed mainly on different membranes, except in Fig. 3b replicates 1 and 3 for H3K4me3 and HA-tag; Fig. 5a replicate 1 for H2A.Z and H3K4me3 (indicated respectively to the probing order). Fig. 5b replicate 1 for H3K122ac and HA-tag, replicate 2 for H3K115ac and HA-tag; Fig. 5b all replicates for H2A.Z and H3.1/3.2.

## Native chromatin immunoprecipitation-sequencing

Native chromatin-immunoprecipitation (ChIP) experiments were performed on MNase-digested nuclear extracts from FH-H3.3A$^{+/-}$ and FH-H3.3A$^{+/-}$ Per1$^{-/-}$, Per2$^{-/-}$ KO nuclei purified from livers collected either every 4 h over 24 h, or at CT8 and CT20. Nuclear extracts were prepared with ~350–530 ug of total protein, according to Khan et al. 2020 and as described above for the IP experiments. DNA quantity in nuclear

extracts was measured on Nanodrop 2000c spectrophotometer (ThermoFisher Scientific). -15–30 ug of chromatin was used per sample for ChIP experiments and 1% of each nuclear extract was saved as input. Inputs from CT0 - CT20 circadian nuclear extracts (related to FH-H3.3A ChIP) and inputs from CT8 and CT20 (related to H3.1/3.2 and BMAL1 ChIP) nuclear extracts were pooled together and stored at −80 °C for further analysis. Nuclear extracts were incubated overnight (18 h) on an end-to-end rotator at 4 °C with 2–4 ug of following antibodies: anti-HA-Tag (C29F4) (Cell Signaling, #3724); anti-Histone H3.1/3.2 (1D4F2) (Active Motif, 61629) and anti-BMAL1 (Bethyl Laboratories, A302-616A). Following overnight incubation with antibodies, equal volume of Protein A and Protein G Dynabeads (Invitrogen) mix was added per sample and nuclear extracts were incubated on an end-to-end rotator for 3–4 h at 4 °C. Unbound fraction was isolated and beads were washed 3 times with 1XD buffer made from 3XD, on rotation at 4 °C for 5 min. After final wash, beads were resuspended in ChIP Elution buffer (1% SDS, 100 mM NaHCO$_3$) and co-immunoprecipitated DNA eluted at 65 °C, with shaking mode at 600 rpm for 15 min. Recovered eluates and inputs were treated with Proteinase K at 10 mg/ml (NEB Molecular Biology Grade, P8107S) at 37 °C for 2h30, followed by DNA purification using QIAquick PCR Purification Kit (QIAGEN). Purified ChIP DNA was stored at -80 °C until further analysis. Concentration of purified DNA samples was assessed on Qubit 4 Fluorometer (Invitrogen) using Qubit dsDNA HS Assay Kit (Invitrogen). Quality of purified DNA samples was assessed on Agilent 4150 TapeStation System using High Sensitivity D5000 ScreenTape assay (Agilent) and analyzed with Agilent TapeStation Software 5.1 prior sequencing.

## Data filtering, ChIP-Seq analysis and graphical representation

Sequencing libraries were prepared with NEBNext UltraII DNA Library Prep Kit for Illumina sequencing, enriching for fragments from 100-150 bp with the 0.9X beads. 150b paired-end (PE) sequencing was performed on Illumina NovaSeq6000 system using NovaSeqS4 flow cell. Approximately 30,000,000 + PE reads were obtained for each replicate (Supplementary Data 3). These were filtered to remove poor quality reads (Q < 20), short reads ( < 75b) and Illumina adapters using cutadapt 4.5, the results were visualized using FastQC 0.12.1. The trimmed reads from each replicate were then mapped separately on the mouse genome GRCm39 with Bowtie2 2.5.4[52] using – very-sensitive −no-mixed−no-discordant and−dovetail options. Samtools 1.18[53], command view -F 1804 -q 20 -f 2, was used to remove poor quality mappings ( < 20), unmapped reads and their mate, non-primary alignments and reads not correctly matched. The resulting alignment was then filtered to remove duplicates and blacklisted regions, following ENCODE recommendations[54], using Picard MarkDuplicates 3.1.1 (Broad Institute 2019) and Bedtools intersect 2.30.0[55] respectively. The alignment of each replicate was then down-sampled to ensure that the same number of sequences were present for each replicate of the same ChIP Seq sample, and thus avoid over-representation of any one of them (Supplementary Data 3). This sub-sampling was carried out using samtools view, and the alignments were sorted using samtools sort. The input samples (ChIP seq control samples) were processed in the same way as the others samples.

In order to compare the alignments obtained for each sample and represent them, deepTools 3.5.5[56] was selected to create a matrix of intensity scores (computeMatrix−beforeRegionStartLength 1000−afterRegionStartLength 1000, combined with plotProfile) from the normalized alignments (bamcoverage−normalizeUsing RPGC−Mnase −maxFragmentLength 200−minFragmentLength 80). These were then plotted either as a heatmap (plotHeatmap) or as an average profile plot (plotProfile). The regions of interest studied in these graphs were the regions of interest were the CLOCK-BMAL1 binding sites recovered from the work of Trott and Menet (2018)[30], the CTCF binding sites from ENCODE project (https://www.ncbi.nlm.nih.gov/geo/query/acc.cgi?acc=GSM918715) and the SINEs class annotation

from UCSC Genome Browser (https://genome-euro.ucsc.edu/). If the sites were not annotated for the mm39 mouse genome, then the coordinates were converted using UCSC genome (https://genome.ucsc.edu/cgi-bin/hgLiftOver).

## Cross-linked ChIP-qPCR

Cross-linked material was directly prepared during the nuclei purification from isolated livers as following. Tissues were grinded in PBS 1X without protease inhibitors. Filtered tissue homogenates were then cross-linked with Formaldehyde solution (Sigma) at 1% final on rotation for 5 min at room temperature, followed by quenching with Glycine at 150 mM final on rotation for 3 min at room temperature. Subsequent nuclei purification steps remained unchanged and done as described in ref. 50, except that Glycine at 150 mM final was added to the sucrose solutions prior ultracentrifugation step. Purified nuclei were stored in Nuclear Lysis Buffer at 1X final with 10% glycerol (NLB 10X stock: 10 mM HEPES-KOH pH 7.6, 100 mM KCl, 0.1 mM EDTA-NaOH pH 8.0) at -80 °C until further use.

To proceed for chromatin preparation, cross-linked nuclei were washed once with 1X Ripperger buffer (Ripperger 10X stock: 200 mM Tris-HCl pH 7.5, 1.5 M NaCl, 20 mM EDTA-NaOH pH 8.0), pelleted at 4000 rpm (1503 g) for 2 min and resuspended again in 1X Ripperger buffer. 1X Ripperger buffer containing 2% SDS was added to the samples and after immediate homogenization, samples were incubated on ice for 30 min, vortexed at low speed every 10 min during the incubation time. Samples were then sonicated 4 times for 3 cycles on Diagenode bioruptor (30 s on/ 30 s off; total 12 cycles) to obtain a smear between 100–300 bp with enrichment for mononucleosomes at 150 bp. In between each 4 cycles, samples were quickly warmed up at 37 °C to dissolve precipitated SDS. Following sonication, samples were centrifuged at 10,000 rpm (9391 g) for 10 min and supernatants kept as cross-linked chromatin for the ChIP experiments, stored at −80 °C until further use. All centrifugation steps were done at 4 °C.

In order to determine DNA concentration prior ChIP experiments, 5% of cross-linked chromatin were decrosslinked in ChIP Elution buffer (1% SDS, 100 mM NaHCO$_3$) for 2 h at 65 °C, treated with Proteinase K at 10 mg/ml (NEB Molecular Biology Grade, P8107S) for 1h30 at 37 °C, and DNA purified using QIAquick PCR Purification Kit (QIAGEN). DNA quantity was measured on Nanodrop 2000c spectrophotometer and sonication efficiency checked on 1.2% agarose gel. For each ChIP, 30 ug of cross-linked chromatin were mixed with 1X Ripperger buffer containing 1% Triton X-100 to get the final concentration of SDS at 0.1% within samples to avoid its precipitation at 4 °C. 1% of each sample was kept as input. ChIP was performed respectively with 2.5 ug and 4 ug of anti-BRG1/SMARCA4 (Bethyl Laboratories, A300-813A) and anti-SMARCA2 antibody [HL1115] (GENETEX, GTX636330), overnight (18 h) on an end-to-end rotator at 4 °C. Following overnight incubation with antibodies, equal volume of Protein A and Protein G Dynabeads (Invitrogen) mix was added per sample and chromatin was incubated on an end-to-end rotator for 3–4 h at 4 °C. Unbound fraction was isolated and beads were first washed 2 times with Wash buffer 1 (0.1% SDS, 1% Triton X-100, 2 mM EDTA-NaOH pH 8.0, 20 mM Tris-HCl pH 8.1, 150 mM NaCl) for 5 min, followed by one wash with Wash buffer 2 (0.1% SDS, 1% Triton X-100, 2 mM EDTA-NaOH pH 8.0, 20 mM Tris-HCl pH 8.1, 500 mM NaCl) for 5 min. All washes were done on rotation at 4 °C. After final wash, beads were resuspended in ChIP Elution buffer (1% SDS, 100 mM NaHCO$_3$) and co-immunoprecipitated DNA eluted at 65 °C, with shaking mode at 600 rpm for 20 min. Recovered eluates and inputs were decross-linked at 65 °C for 2 h and treated with Proteinase K at 10 mg/ml for 1h30 at 37 °C, followed by DNA purification using QIAquick PCR Purification Kit. Purified ChIP DNA was stored at -80 °C until further analysis.

Real-time qPCR reactions were performed as follows. Each sample was deposited in three wells as technical replicates on Hard-Shell PCR 96-well thin wall plates (Bio-Rad). Reaction master mixes were prepared using ITaq Universal SYBR Green Supermix (Bio-Rad) and specific forward/ reverse primer sets. RT-qPCR were performed with CFX96 Real-Time PCR Detection System (Bio-Rad) and analyzed by CFX Maestro Software (version 2.3). Enrichment at clock-gene E-boxes (*Per1*, *Dbp*, *Per2*, *Reverba*, *Cry2*, *Rorc*) was assessed, calculated as percentage of the immunoprecipitated input. All the results are represented as the average ± s.e.m of three independent biological replicates. Primers for these regions[57] were synthesized by Eurogentec; primer sequences are available in the Supplementary Data 4.

## mRNA expression levels and RT-qPCR

Whole liver extracts from FH-H3.3A$^{+/-}$ and FH-H3.3A$^{+/-}$ *Per1*$^{-/-}$, *Per2*$^{-/-}$ KO livers collected at CT8 and CT20 were obtained during the nuclei purification as follows. Tissues were grinded in PBS 1X containing protease inhibitors and aliquots of tissue homogenates were stored at −80 °C until further use. Total RNA was extracted from whole liver extracts with TRIzol LS Reagent (Invitrogen) according to manufacturer's instructions and quantified on Nanodrop 2000c spectrophotometer. cDNA was prepared from 2 ug of purified RNA using IScript cDNA Synthesis kit (Bio-Rad) according to manufacturer's instructions. Real-time qPCR reactions were performed as follows. Each sample was deposited in two to three wells as technical replicates on Hard-Shell PCR 96-well thin wall plates (Bio-Rad). Reaction master mixes were prepared using ITaq Universal SYBR Green Supermix (Bio-Rad) and specific forward/ reverse primer sets. RT-qPCR were performed with CFX96 Real-Time PCR Detection System (Bio-Rad) and analyzed by CFX Maestro Software (version 2.3). All the results are represented as the average ± s.e.m. of three independent biological replicates of relative expression calculated with the 2$^{-\Delta\Delta Ct}$ method and normalized to *Rps9* cDNA levels. Primer sequences were mainly obtained from the Harvard PrimerBank database and synthesized by Eurogentec; they are available in the Supplementary Data 4.

## Real-time bioluminescence monitoring assay of circadian oscillations

FLAG-HA-H3.3B f/f animals were crossed to H3.3A f/f strain[58] and embryos were sacrificed at E14 to generate mouse embryonic fibroblasts. Cell lines were cultured at 37 °C with a 5% CO$_2$ atmosphere in DMEM-high glucose medium (Sigma-Aldrich) supplemented with 10% FBS (Gibco) and 1% penicillin-streptomycin at 25 units/ml (ThermoFisher).

Circadian bioluminescence rhythms were monitored on confluent wild-type and tamoxifen-treated double-floxed FH-H3.3A$^{-/-}$, FH-H3.3B$^{-/-}$ ERT2-Cre Bmal1: luciferase reporter mouse embryonic fibroblasts. To obtain H3.3 knock-out, cells were treated with 4-OHT, cis/trans-4-hydroxytamoxifen (Sigma-Aldrich) at 5 µM for 3.5 days and knock-out efficiency was verified by western-blotting. Both WT and KO cells were seeded at high confluency in a black 24 well plate with flat and clear bottom for high-throughput microscopy (Ibidi) and cells were synchronized with Dexamethasone (Sigma-Aldrich) at 100 nM for 1h30. After synchronization, cell culture medium was replaced, supplemented with HEPES (Gibco) at 10 mM, sodium bicarbonate solution (Gibco) at 1% and 200 µM of D-Luciferin Sodium Salt (PJK GmbH) and bioluminescence was monitored with LumiCycle 96 luminometer for circadian biology (ACTIMETRICS). LumiCycle Data Analysis software was used to visualize the data, to perform detrending analysis and calculation of oscillation period and amplitude.

## Subcellular protein tissue fractionation

Nucleoplasm and chromatin fractions were extracted from FH-H3.3A$^{+/-}$ and FH-H3.3A$^{+/-}$ *Per1*$^{-/-}$, *Per2*$^{-/-}$ KO liver pieces collected at CT8 and CT20, using Subcellular Protein Fractionation Kit for Tissues (Thermo Scientific) according to the manufacturer's instructions. Whole liver lysates were obtained by incubating small liver pieces from the same tissues as for the fractionation in RIPA lysis buffer for 30 min on ice,

then sonicated for 6 cycles on Diagenode bioruptor (30 s on/ 30 s off) at 4 °C, followed by centrifugation at 14,000 rpm for 10 min. Proteins from all fractions were quantified using DC protein assay.

### MG132 treatment and subcellular protein fractionation

Unsynchronized confluent murine wild-type and $Per1^{-/-}$, $Per2^{-/-}$, $Per3^{-/-}$ triple-knockout (PerTKO) lung fibroblasts were treated with MG132 proteasome inhibitor (Sigma-Aldrich) for 4 h prior chromatin extraction. Subcellular protein fractionation was performed using Subcellular Protein Fractionation Kit for Cultured Cells (Thermo Scientific) according to the manufacturer's instructions. Whole cell lysates were obtained by incubating a part of the same cell pellets that were used for the fractionation, in RIPA lysis buffer for 20 min on ice, then sonicated for 6 cycles on Diagenode bioruptor (30 s on/ 30 s off) at 4 °C, followed by centrifugation at 14,000 rpm for 10 min. Proteins from all fractions were quantified using DC protein assay.

### Reanalysis of published siRNA screens

To verify whether specific PBAF and cBAF complex components have impact on circadian clock regulation, we searched and reanalyzed data from the previously published genome-wide siRNA screen for modifiers of the circadian clock[45]. Data from all available replicates for the siRNAs of target genes, $Cry2$ positive and negative controls were taken into account from all plates. Technical duplicates for given control siRNAs were averaged to obtain a single value and thus in total four traces for each siRNA, corresponding to two averaged duplicates from each plate. For the siRNAs of target genes, since one technical duplicate was available on each plate, obtained values were kept without calculating the average and the four traces correspond to individual technical replicates. Statistical analysis for period length was performed with available data from the published study (see NIHMS144235-supplement-01 file[45] for period distribution for negative and positive control siRNA, respectively 25.20 h ± 0.55 for and 29.43 h ± 0.65; BioGPS centralized gene-annotation portal[59] with Circadian Layout plugin−Circadian Genomics Screen Database - BioGPS - your Gene Portal System for respective target siRNA made available by the authors of the genome wide RNAi screen[45]). For the negative and positive control siRNA and $Phf10$ screen, data from only two replicates were available in the database. In order to perform statistical analysis for all replicates of all target genes, the duplicates of negative control siRNA were taken twice into account for the t-test analysis.

### Impact of H3.3 loss on clock gene expression

To assess the impact of H3.3 loss on core-clock and clock output gene expression, we collected non-synchronized H3.3 wild-type (untreated) and H3.3DKO (tamoxifen-treated) MEFs 6.5 d post-treatment. Total RNA was extracted from cell pellets with TRIzol LS Reagent (Invitrogen) according to manufacturer's instructions and quantified on Nanodrop 2000c spectrophotometer. cDNA was prepared from 2 ug of purified RNA using IScript cDNA Synthesis kit (Bio-Rad) according to manufacturer's instructions. Real-time qPCR reactions were performed as follows. Each sample was deposited in three wells as technical replicates on Hard-Shell PCR 96-well thin wall plates (Bio-Rad). Reaction master mixes were prepared using ITaq Universal SYBR Green Supermix (Bio-Rad) and specific forward/ reverse primer sets. RT-qPCR were performed with CFX96 Real-Time PCR Detection System (Bio-Rad) and analyzed by CFX Maestro Software (version 2.3). All the results are represented as the average ± s.e.m. of three independent biological replicates of relative expression calculated with the $2^{-\Delta\Delta Ct}$ method and normalized to $Rps9$ cDNA levels. Primer sequences were mainly obtained from the Harvard PrimerBank database and synthesized by Eurogentec; they are available in the Supplementary Data 4.

### Impact of PBAF-cBAF specific complex components on clock gene expression

In order to assess the impact of PBAF-cBAF specific components on core-clock and clock output gene expression, we performed siRNA-mediated knockdown of $Pbrm1$, $Arid2$, $Brd7$, $Phf10$, $Smarca4$ ($Brg1$), $Smarca2$ ($Brm$) and $Arid1b$ in U2OS Bmal1:Luciferase cell line. 30-50% confluent non-synchronized cells were transfected with either siRNA Negative Control, or a specific siRNA targeting a given PBAF/cBAF complex component, using INTERFERin® (Polyplus) transfection reagent according to the manufacturer's instructions. siRNA duplex sequences were provided and synthetized by Eurogentec on request. Total RNA was extracted from cell pellets with TRIzol LS Reagent (Invitrogen) according to manufacturer's instructions and quantified on Nanodrop 2000c spectrophotometer. cDNA was prepared from 2 ug of purified RNA using IScript cDNA Synthesis kit (Bio-Rad) according to manufacturer's instructions. Real-time qPCR reactions were performed as follows. Each sample was deposited in three wells as technical replicates on Hard-Shell PCR 96-well thin wall plates (Bio-Rad). Reaction master mixes were prepared using ITaq Universal SYBR Green Supermix (Bio-Rad) and specific forward/ reverse primer sets. RT-qPCR were performed with CFX96 Real-Time PCR Detection System (Bio-Rad) and analyzed by CFX Maestro Software (version 2.3). All the results are represented as the average ± s.e.m. of three independent biological replicates of relative expression calculated with the $2^{-\Delta\Delta Ct}$ method and normalized to $Rps9$ cDNA levels. Primer sequences were mainly obtained from the Harvard PrimerBank database and synthesized by Eurogentec; they are available in the Supplementary Data 4. Part of the same cell pellets was kept to assess the knockdown efficiency by western-blotting; whole cell extracts in RIPA buffer were obtained as described above.

### Reporting summary

Further information on research design is available in the Nature Portfolio Reporting Summary linked to this article.

## Data availability

The ChIP-seq data generated in this study have been deposited in NCBI's Gene Expression Omnibus and are accessible through GEO Series accession number GSE284165. The mass spectrometry proteomics data generated in this study have been deposited in the ProteomeXchange Consortium via the PRIDE partner repository database under the dataset identifier PXD061974. Source data are provided with this paper.

## Code availability

All scripts are available at (https://github.com/Thrylia/ChIP_HA-Complexe)[60].

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

## Acknowledgements

The work was supported by an ATIP Avenir installation grant, ANR (ANR–20-CE12-0013), Pack Ambition Research AURA (2000671901) and a EU H2020 (101016726) to KP. DLs research was supported by a PhD fellowship from the doctoral school ED 340 BMIC, ENS de Lyon and the "Fondation ARC pour la recherche sur le cancer" 4th year PhD fellowship. YS was supported by Adopting Sustainable Partnerships for Innovative Research Ecosystem (ASPIRE), AMED, JAPAN (JP25jf0126017 and JP23jf0126003) grants. We would like to thank Stefan Dimitrov for kindly providing us the H3.3 mouse models. We would like to especially thank Aysegul Ors for validating the mouse lines, genotyping protocols, establishing initial protocol conditions for H3.3 analysis and for help in preliminary work suggesting H3.3 dynamics at specific clock loci. Edwige Belotti for valuable advice for mouse handling at the ENS Lyon and genotyping protocols. We would like to also thank Marie-Paule Felder Schmittbuhl and Urs Albrecht for *Per1*−/−; *Per2*  *Brdm*−/− mice. We acknowledge the contribution of SFR Biosciences (UMS3444/CNRS, US8/Inserm, ENS de Lyon, UCBL) facility: Adeline Page and Frédéric Delolme from PSF for the mass spectrometry analysis, Nadine Aguilera from PBES for animal husbandry. Kazumi Abe and Shota Takashima from the Department of Computational Biology and Medical Sciences of the University of Tokyo for their help with sequencing. The bioinformatics analyses were performed on the Core Cluster of the Institut Français de Bioinformatique (IFB) (ANR-11-INBS-0013). We would like to thank Benjamin Gillet, Sandrine Hughes and Yves Dusabyinema at the sequencing platform at IGFL. We also thank Jean-Christophe Andrau, Benjamin Loppin, Geneviève Fourel, Jerome Menet, Peter Becker and members of the French chromatin community (GDR ADN&G) for feedback.

## Author contributions

K.P. conceived and directed the project with D.L. D.L. conducted the experiments with help from D.S. for animal experimentation and M.S., Y.S. for ChIP-sequencing experiments. A.P. performed all the bioinformatics analyses. K.P. and D.L. wrote the manuscript with input from all co-authors.

## Competing interests

The authors declare no competing interests
