## [Transparent Peer Review file · Nature Communications]

PBAF/cBAF reorganization on H3.3 chromatin regulates BMAL1 activity in the absence of circadian negative feedback

Corresponding Author: Mr Kiran Padmanabhan

Version 0:

Reviewer comments:

Reviewer #1

(Remarks to the Author)

This study by Letkova and colleagues discovered that H3.3 deposition in the liver chromatin exhibits a circadian pattern, peaking during the subjective day. This is accompanied by rhythmic recruitment of BMAL1/CLOCK and perhaps PBAF and BRG1/BAF complexes. On the other hand, Per1/Per2 deficiency results in a reduction of BMAL1 recruitment to the H3.3 nucleosome, as well as disassembly of the PBAF/BRG1/BAF complexes likely due to decreased nuclear level of various components of these complexes. Overall, this work is a technical tour de force, unveiling the detailed mechanism underlying circadian transcriptional regulation by BMAL1/CLOCK. With that being said, there are also several prominent issues that should be addressed before this manuscript can be accepted for publication.

Major Concerns:

1. The authors propose the notion of "a remodeler switch from PBAF to BRM/cBAF" in the absence of Per, but I do not think this phenomenon should be referred to as a "switch". It is simply that PBAF is disrupted by Per deficiency while BRM/cBAF is not. Moreover, I do not find this so-called "switch" to be very insightful. In my opinion, the observation of rhythmic H3.3 deposition and cyclic recruitment of BMAL1 and PBAF/BRG1/BAF complexes to these H3.3 nucleosomes, which are disrupted in Per KO are more interesting and valuable findings. How does lack of Per lead to reduction of PBAF components (ARID2, BRG1, BRD7 and PHF10) in the nucleus? Are these effects due to clock disturbance in general or specific to Per deficiency? Would lack of Bmal1 or Cry lead to similar changes?
2. Loss of SWI/SNF subunits does not substantially alter circadian rhythm in cultured cells, which could be due to genetic redundancy and/or the robustness of the circadian clock per se. Does loss of PBAF/BRG1/BAF components affect clock gene transcription? What about loss of H3.3? It would be good to see some effects on gene expression as a consequence of SWI/SNF deficiency to demonstrate that these complexes are indeed involved in circadian transcriptional regulation.
3. In Fig.4d, BRM binding at E-box elements clearly show a trend of decrease in Per KO, although the differences do not reach statistical significance (likely due to the small sample size).

Minor concerns:

1. On several immunoblots, there are multiple bands (such as those for BRG1, BRD7 and PHF10). The authors should indicate the correct band on the blot.
2. The arrows in Fig.5d are quite confusing. The authors should edit this model figure to make it more self-explanatory.
3. It would be helpful to include some introduction regarding the different SWI/SNF complexes in the Intro rather than the Discussion.

Reviewer #2

(Remarks to the Author)

Letkova and colleagues describe in this article that BMAL1/CLOCK is recruited to H3.3 nucleosomes during the peak of BMAL1 activity, associated with the PBAF remodeler complex. This mechanism is dependent of PERs proteins. This provides new information about the molecular mechanisms involve in the regulation of transcription by BMAL1. This is

globally an interesting and well performed and written article, but a few corrections are required before publication.

- While CRY directly binds to BMAL1 and acts as its main repressor, authors did not study if and how CRY is recruited to chromatin along the PBAF complex. This is an important component of the mechanism and additional information is required.

- Many conclusions are not supported by quantification and statistical analysis. This must be added. Figure S8 completely lacks quantitative analysis.

- Globally, the supplementary tables are poorly informative. Additional tables must provide quantitative values and statistical analysis. This is particularly critical for Table S1. In addition, both proteomics and ChIP-seq data must be deposited in recognized databases with an access provided to reviewers and future readers.

Reviewer #3

(Remarks to the Author)

Binding of the circadian transcription factor (TF) CLOCK-BMAL1 to DNA is associated with daily rhythms of chromatin remodeling and nucleosome eviction. In this paper submitted by Letkova and collaborators, the authors used various approaches to uncover the underlying mechanisms, focusing on the histone variant H3.3. and its nucleosomal incorporation into fragile nucleosomes. They took advantage of a mouse strain carrying a tagged H3.3 allele to perform multiple biochemical and genomics assays at the peak/trough of CLOCK-BMAL1-mediated transcription or across the circadian cycle. They found that H3.3 incorporation at CLOCK:BMAL1 cis-regulatory elements (CREs) is rhythmic and peaks during the day, coincidentally to CLOCK-BMAL1 DNA binding. CLOCK-BMAL1 daytime binding to H3.3 nucleosomes is associated with rhythmic recruitment of members of the PBAF complex and BRG1. The authors also showed that ARID2, which is essential for the formation of PBAF complex, is expressed at very low levels in Per1/2 DKO liver, leading to a switch in chromatin remodeling complexes from PBAF/BRG1 to cBAF/BRM.

This is overall a well-conducted study that sheds light on interesting and important mechanisms that underlie daily rhythms in chromatin remodeling and circadian transcription. The biochemical and genomics assays are well-conducted and thorough. The results are compelling and relevant for the circadian field. My main comment, that I would like the authors to address in their revised manuscript, stems from how the authors interpret their findings with PerKO mice and relate it to circadian transcription in wild-type mice. While CLOCK-BMAL1 activity in PerKO mice is expected to resemble that of WT CT8 due to the lack of circadian repression, data clearly show that it is not the case because of ARID2 depletion in PerKO mice which creates a broad remodeler switch from PBAF/BRG1 to cBAF/BRM. Moreover, CLOCK-BMAL1 binding to H3.3 nucleosomes is completely different between WT CT8 and PerKO. Thus, data from PerKO mice cannot inform much about circadian repression, and sentences in the manuscript like "decrease of ARID2 and BRG1 in PerKO livers disrupts the PBAF and cBAF/BRG1 complexes on chromatin, highlighting a critical role for these SWI/SNF remodelers in circadian negative feedback." (lines 195-198) need to be edited/corrected.

I would suggest that the authors rather use the PerKO data as a "tool" to uncover new mechanisms related to CLOCK:BMAL1 activity in wild-type mice, and state it as such in their paper. Comparison between WT CT8 and PerKO indeed indicates that in WT mice, CLOCK:BMAL1 associates preferentially to PBAF and not cBAF, which is a novel and important finding on its own. The low levels of ARID2/BRG1 in PerKO mice depletes PBAF/BRG1 complexes, leading to increased interaction between CLOCK:BMAL1 and cBAF, increased H3.3 nucleosome incorporation, yet, decreased interaction of CLOCK:BMAL1 with H3.3 nucleosomes. I understand that this may be what the authors intended, but this is not as clear in the current version of the manuscript.

Other major comments:

1. It is unclear why the section about IP/MS of H3.3A nucleosomes (lines 72-82) only includes data from the comparison between WT CT20 and PerKO, and is only shown as supplementary data. Are there any differences between WT CT8 and PerKO, when CLOCK-BMAL1 is not repressed for both conditions? Can the authors comment on that? And show all the data?

Also, the rationale of this experiment is unclear as one may not expect major differences between the samples because only a small fraction of cis-regulatory regions (containing H3.3) is bound by CLOCK:BMAL1. If an effect is only observed for PerKO (i.e., no difference between WT CT8 and WT CT20), then, the effects may be more related to general effects of Per1/2 KO on the stability of PBAF complex components and/or assembly of the PBAF complex. This would align with results shown in Fig. 3, where ARID2 levels are down in PerKO.

2. A relevant control for results shown in Fig. 1D would have been all TSS and/or all TSS not bound by CLOCK:BMAL1. Also, regarding the sentence lines 120-121 ("The time of enrichment of H3.3 at clock genes coincided with the peak recruitment of CLOCK-BMAL1 and active RNA Polymerase II"), it seems that H3.3 enrichment precedes CLOCK-BMAL1 binding by a few hours, since levels are down at CT8. Moreover, the authors may want to normalize their H3.3 signal to nucleosome signals at each time point, since nucleosome signal is also rhythmic and low at time of strong CLOCK-BMAL1 DNA binding.

3. Results for Fig. 2 should be clarified. Is the effect specific to CLOCK-BMAL1 targets, or for all genes. The authors' finding that H3.3 is enriched at both CLOCK:BMAL1 sites and CTCF sites suggest that the effects in PerKO are general, see comment 1 as well.

4. Section on BRG1 and BRM ChIP-qPCR (Fig. 4). The higher recruitment of BRG1 at CT20 vs. CT8 seems opposite to

expectations, yet not discuss. In fact, it seems that the recruitment of BRG1 and BRM are inversely correlated to H3.3 nucleosome signal. The authors should clarify this result, and whether it is in line with their models/hypotheses and/or the literature.

5. Can the authors elaborate on why CLOCK:BMAL1 fail to bind fragile H3.3 nucleosomes in Per KO mice?

Minor comments:

1. Lines 69-71: "Remarkably, H3.3A complexes retained H2A.Z but did not include canonical H3.1/2 at the two time points and in PerKO animals, CLOCK-BMAL1 was no longer enriched on variant H3.3 nucleosomes." I would describe the results in PerKO in a separate sentence, as the current reading is confusing (i.e., H3.1/2 is not included at the two time points and in PerKO animals, and CLOCK-BMAL1 was no longer enriched in H3.3 animals in PerKO).

Version 1:

Reviewer comments:

Reviewer #1

(Remarks to the Author)

The authors have addressed all my concerns.

Reviewer #2

(Remarks to the Author)

Authors have adequately answered reviewers' queries and this interesting article is now acceptable for publication.

Reviewer #3

(Remarks to the Author)

The authors appropriately addressed my comments as well as those from the two other reviewers. I just note that a correction made by the authors led Fig. 2d to the referenced before other Fig. 2 panels, which needs to be corrected (see line 121). This manuscript is suitable for publication in Nature Communication.

Please find a rebuttal to the reviewer comments below.

Our responses are in blue.

Reviewer #1 (Remarks to the Author):

This study by Letkova and colleagues discovered that H3.3 deposition in the liver chromatin exhibits a circadian pattern, peaking during the subjective day. This is accompanied by rhythmic recruitment of BMAL1/CLOCK and perhaps PBAF and BRG1/BAF complexes. On the other hand, *Per1/Per2* deficiency results in a reduction of BMAL1 recruitment to the H3.3 nucleosome, as well as disassembly of the PBAF/BRG1/BAF complexes likely due to decreased nuclear level of various components of these complexes. Overall, this work is a technical tour de force, unveiling the detailed mechanism underlying circadian transcriptional regulation by BMAL1/CLOCK. With that being said, there are also several prominent issues that should be addressed before this manuscript can be accepted for publication.

We thank the reviewer for their thoughtful and appreciative remarks.

Major Concerns:

1. The authors propose the notion of "a remodeler switch from PBAF to BRM/cBAF" in the absence of *Per*, but I do not think this phenomenon should be referred to as a "switch". It is simply that PBAF is disrupted by *Per* deficiency while BRM/cBAF is not. Moreover, I do not find this so-called "switch" to be very insightful. In my opinion, the observation of rhythmic H3.3 deposition and cyclic recruitment of BMAL1 and PBAF/BRG1/BAF complexes to these H3.3 nucleosomes, which are disrupted in *Per* KO are more interesting and valuable findings.

We agree with the reviewers point, replacing the concept of a "remodeler switch" with "remodeler reorganization" and have modified the title of the manuscript to reflect our findings perhaps better and also highlight the role of H3.3 in the clock, which is indeed a valuable and novel finding of our work.

The title now reads: "Reorganization of PBAF/cBAF remodelers on H3.3 chromatin regulates BMAL1 activity in the absence of circadian negative feedback".

How does lack of *Per* lead to reduction of PBAF components (ARID2, BRG1, BRD7 and PHF10) in the nucleus? Are these effects due to clock disturbance in general or specific to *Per* deficiency? Would lack of *Bmal1* or *Cry* lead to similar changes?

We measured the relative expression of PBAF components in the *Per*KO livers (Fig.3c) and found that in all cases transcript levels did not differ between wildtype and knockout livers. We also compared expression of all cBAF/PBAF remodeler components in *Bmal1*^{-/-} and *Cry1*^{-/-}; *Cry2*^{-/-} knockout liver tissues from published datasets (<https://doi.org/10.1073/pnas.2015803118>) (Figure A, below). The expression of most components was not altered significantly in the clock mutant mice.

Figure A. Differential expression of cBAF/PBAF components in *Bmal1* and *Cry1; Cry2* double knockout mice.

We next checked if the protein levels were altered in *Per1-3* triple knockout (PerTKO) and *Bmal1*^{-/-} fibroblasts, which would allow us to explore changes in protein stability. Strikingly, the loss of ARID2 that we observed in livers was not seen in cultured cells likely suggesting different regulatory mechanisms operating in post-mitotic cells (Figure B, below; ARID2 and BMAL1 protein relative quantity was normalized to H3).

Figure B.

Similar phenomena have been described in cancer cells, where the loss of one remodeler component can lead to a loss of other proteins within the complex.

In melanoma lines that exhibit a complete loss of ARID2 (Carcamo et al., 2022), a concomitant reduction in protein levels of PBAF-specific members was observed. Similar phenomena have been attributed to a deficiency in PBAF subcomplex assembly in a few other manuscripts (Mashtalir et al., 2018; Schick et al., 2019; Yan et al., 2005).

Thus, a direct mechanism for decrease in PBAF components is probably multi-factorial and unclear at the moment and shedding light on circadian control of this mechanism, we believe, goes beyond the scope of our current manuscript.

2. Loss of SWI/SNF subunits does not substantially alter circadian rhythm in cultured cells, which could be due to genetic redundancy and/or the robustness of the circadian clock per se. Does loss of PBAF/BRG1/BAF components affect clock gene transcription? What about loss of H3.3? It would be good to see some effects on gene expression as a consequence of SWI/SNF deficiency to demonstrate that these complexes are indeed involved in circadian transcriptional regulation.

To determine if loss of PBAF/BRG1/BAF components affect clock gene transcription, we analyzed publicly available datasets for liver knockout models of BAF/PBAF components for the GO term + KEGG pathway 'circadian rhythms'.

Arid1a/b liver-specific double knockout (<https://doi.org/10.1038/s43018-020-00109-0>) shows several core- and clock-output genes being affected (mainly downregulated) (Figure C, below).

Arid1a/b double knock-out liver DESeq2

Figure C.

Analysis of **Arid2 liver-specific knockout** (<https://doi.org/10.1038/s41418-022-01090-0>) showed no difference. Indeed, in this dataset, surprisingly only 4 genes were differentially regulated (as stated by the authors in their publication, and the data could not be retrieved).

For remaining specific PBAF-BRG1/cBAF components, no liver-specific datasets were available. We therefore analyzed the available datasets from cancer cell lines that were knocked down for Pbrm1, Arid2, Brd7, Smarca4 and Smarca2 (GSE108388) (Figure D, below).

Figure D.

However, there were no significant differences in clock gene expression suggesting that the knockout of Arid1 seems to be the only remodeler component with a direct impact on the liver tissue clock.

We performed siRNA-mediated knockdowns of specific PBAF-BRG1/cBAF and BRM/cBAF complex components in a U2OS cell line to assess their impact on clock gene expression. We found that knockdowns of these specific components had minor effects on clock gene expression, the strongest impact being observed upon *Arid1b*, *Brg1* (*Smarca4*) and *Brd7* knockdowns, with downregulated expression of *Dbp*, *Per1* and *Per2* genes. These results are now integrated in the Supplementary figure S9a-h (and are shown below as well).

To determine the impact of H3.3 on clock gene expression beyond the change in the circadian period in real-time bioluminescence assays, we measured gene expression in H3.3 DKO fibroblasts. Many of the clock genes were downregulated as is expected from the loss of H3.3 that has been shown to facilitate active gene transcription. These results are now integrated in the Supplementary Figure S7a (and are shown below as well).

3. In Fig.4d, BRM binding at E-box elements clearly show a trend of decrease in *Per* KO, although the differences do not reach statistical significance (likely due to the small sample size).

We performed two more additional BRM ChIP-qPCR experiments. However, plotting all five replicates on the same graph, or even four out of the five (while removing the outlier from the original third replicate) shows no statistical significant difference between the three conditions. These results are now integrated in the Fig.4d with all five replicates (and are shown below as well).

Minor concerns:

1. On several immunoblots, there are multiple bands (such as those for BRG1, BRD7 and PHF10). The authors should indicate the correct band on the blot.

Many of the BAF/PBAF complex components have multiple coding isoforms. Nevertheless, to limit confusion among readers, we cropped the images to show the primary isoform. Entire uncropped images for the blots can still be found in extended data files.

2. The arrows in Fig.5d are quite confusing. The authors should edit this model figure to make it more self-explanatory.

We have edited the figure further to simplify the message.

3. It would be helpful to include some introduction regarding the different SWI/SNF complexes in the Intro rather than the Discussion.

We agree with the referee and have added a section to the introduction (lines 18-27).

Reviewer #2 (Remarks to the Author):

Letskova and colleagues describe in this article that BMAL1/CLOCK is recruited to H3.3 nucleosomes during the peak of BMAL1 activity, associated with the PBAF remodeler complex. This mechanism is dependent of PERs proteins. This provides new information about the molecular mechanisms involve in the regulation of transcription by BMAL1. This is globally an interesting and well performed and written article, but a few corrections are required before publication.

We appreciate the comments of this referee very much.

- While CRY directly binds to BMAL1 and acts as its main repressor, authors did not study if and how CRY is recruited to chromatin along the PBAF complex. This is an important component of the mechanism and additional information is required.

During our initial analysis, we probed both endogenous PBAF and FLAG-HA epitope tagged H3.3 complexes (from Figure 1 and Figure 3) for the presence of both CRYs and PERs. We did not detect either repressor component in the H3.3/ endogenous PBAF complexes. So, the H3.3-remodeler complexes very likely reflect the active form of the CLOCK-BMAL1 transcription factor complex.

- Many conclusions are not supported by quantification and statistical analysis. This must be added. Figure S8 completely lacks quantitative analysis.

We apologize for the oversight. We have now quantified the blots in Figures comparing Wildtype to the PerKO and as well H3.3 protein levels after tamoxifen treatment, HIRA and DAXX protein levels. These include the quantification of data shown in Figures 1a,1f, 3a, 3b, 4a, 4b, 5a, 5b, S5a, S6b and these have been provided as an extended data file. We have also quantified and performed statistical analysis on panels in Figure S10a and included it in the panel.

- Globally, the supplementary tables are poorly informative. Additional tables must provide quantitative values and statistical analysis. This is particularly critical for Table S1. In addition, both proteomics and ChIP-seq data must be deposited in recognized databases with an access provided to reviewers and future readers.

Quantitative values as well as adj. p-values are included in the Table S1. We highlighted as well peptides that are significantly missing in the H3.3 interactome in the PerKO livers.

Earlier this year, (March 2025) we deposited the ChIP-seq data at GEO, GSE284165 and all the scripts are available at https://github.com/Thrylia/ChIP_HA-Complex. The mass spectrometry proteomics data were also deposited to the ProteomeXchange Consortium via the PRIDE partner repository with the dataset identifier PXD061974 and 10.6019/PXD061974.

Reviewer #3 (Remarks to the Author):

Binding of the circadian transcription factor (TF) CLOCK-BMAL1 to DNA is associated with daily rhythms of chromatin remodeling and nucleosome eviction. In this paper submitted by Letkova and collaborators, the authors used various approaches to uncover the underlying mechanisms, focusing on the histone variant H3.3. and its nucleosomal incorporation into fragile nucleosomes. They took advantage of a mouse strain carrying a tagged H3.3 allele to perform multiple biochemical and genomics assays at the peak/trough of CLOCK-BMAL1-mediated transcription or across the circadian cycle. They found that H3.3 incorporation at CLOCK:BMAL1 cis-regulatory elements (CREs) is rhythmic and peaks during the day, coincidentally to CLOCK-BMAL1 DNA binding. CLOCK-BMAL1 daytime binding to H3.3 nucleosomes is associated with rhythmic recruitment of members of the PBAF complex and BRG1. The authors also showed that ARID2, which is essential for the formation of PBAF complex, is expressed at very low levels in Per1/2 DKO liver, leading to a switch in chromatin remodeling complexes from PBAF/BRG1 to cBAF/BRM.

This is overall a well-conducted study that sheds light on interesting and important mechanisms that underlie daily rhythms in chromatin remodeling and circadian transcription. The biochemical and genomics assays are well-conducted and thorough. The results are compelling and relevant for the circadian field.

We thank the referee for this thoughtful review. We have addressed the points raised below.

My main comment, that I would like the authors to address in their revised manuscript, stems from how the authors interpret their findings with PerKO mice and relate it to circadian transcription in wild-type mice. While CLOCK-BMAL1 activity in PerKO mice is expected to resemble that of WT CT8 due to the lack of circadian repression, data clearly show that it is not the case because of ARID2 depletion in PerKO mice which creates a broad remodeler switch from PBAF/BRG1 to cBAF/BRM. Moreover, CLOCK-BMAL1 binding to H3.3 nucleosomes is completely different between WT CT8 and PerKO. Thus, data from PerKO mice cannot inform much about circadian repression, and sentences in the manuscript like “decrease of ARID2 and BRG1 in PerKO livers disrupts the PBAF and cBAF/BRG1 complexes on chromatin, highlighting a critical role for these SWI/SNF remodelers in circadian negative feedback.” (lines 195-198) need to be edited/corrected.

We agree with the referee and we have deleted the highlighted sentence, which was speculative. And modified the text in the various sections including the discussion. These are now highlighted in blue.

I would suggest that the authors rather use the PerKO data as a “tool” to uncover new mechanisms related to CLOCK:BMAL1 activity in wild-type mice, and state it as such in their paper. Comparison between WT CT8 and PerKO indeed indicates that in WT mice, CLOCK:BMAL1 associates preferentially to PBAF and not cBAF, which is a novel and important finding on its own. The low levels of ARID2/BRG1 in PerKO mice depletes PBAF/BRG1 complexes, leading to increased interaction between CLOCK:BMAL1 and cBAF, increased H3.3 nucleosome incorporation, yet, decreased interaction of CLOCK:BMAL1 with H3.3 nucleosomes. I understand that this may be what the authors intended, but this is not as clear in the current version of the manuscript.

We thank the referee for clarifying and indeed summarizing the main thrust of our manuscript and have modified the discussion to reflect this point more clearly (lines 340-346).

Other major comments:

1. It is unclear why the section about IP/MS of H3.3A nucleosomes (lines 72-82) only includes data from the comparison between WT CT20 and PerKO, and is only shown as supplementary data. Are there any differences between WT CT8 and PerKO, when CLOCK-BMAL1 is not repressed for both conditions? Can the authors comment on that? And show all the data?

We used the mass spectrometry analysis primarily as a guide for our experiments and performed only 2 replicates and therefore we relegated the information to Supplementary data files. We did not expect significant changes for remodelers in binding H3.3 nucleosomes day vs night. Our analysis showed that a few peptides were different for BAF/PBAF complex components between CT8 and CT20. We

have added in Table S1 the comparison between CT8 vs PerKO highlighting in green the handful of peptides that are altered between the conditions, with the relevant adj p-values.

Also, the rationale of this experiment is unclear as one may not expect major differences between the samples because only a small fraction of cis-regulatory regions (containing H3.3) is bound by CLOCK:BMAL1. If an effect is only observed for PerKO (i.e., no difference between WT CT8 and WT CT20), then, the effects may be more related to general effects of Per1/2 KO on the stability of PBAF complex components and/or assembly of the PBAF complex. This would align with results shown in Fig. 3, where ARID2 levels are down in PerKO.

The rationale for the mass spec experiments was to potentially identify deposition chaperones that might account for H3.3 dynamics that we had observed by preliminary ChIP-qPCR experiments. As stated above, only a few peptides were altered between CT8 and CT20 (highlighted in Table S1) for BAF/PBAF complexes and none of the known chaperones for H3.3 were either not reliably detected or were not found to differ (Supplementary Figure S6b).

We agree with the referee that the difference in the assembly or stability of PBAF complex components is therefore a general effect we observe in PerKO tissues.

2. A relevant control for results shown in Fig. 1D would have been all TSS and/or all TSS not bound by CLOCK:BMAL1.

Thank you for pointing out that we missed reporting this important control. We analyzed the data and have included heat map panels in the main figure file as Fig.2d (shown as well below). As is clear, the H3.3 variant is preferentially enriched at CLOCK:BMAL1 genes.

Also, regarding the sentence lines

120-121 (“The time of enrichment of H3.3 at clock genes coincided with the peak recruitment of CLOCK-BMAL1 and active RNA Polymerase II”), it seems that H3.3 enrichment precedes CLOCK-BMAL1 binding by a few hours, since levels are down at CT8.

We agree with this and changed the text to reflect the point raised by the referee. The text now reads:

Lines 127-129: “The time of enrichment of H3.3 at clock genes preceded the peak recruitment of CLOCK-BMAL1 and active RNA Polymerase II”

Moreover, the authors may want to normalize their H3.3 signal to nucleosome signals at each time point, since nucleosome signal is also rhythmic and low at time of strong CLOCK-BMAL1 DNA binding.

We thank the referee for this comment. We normalized the H3.3 ChIP-Seq signal to the pooled nucleosome input (across all time points). The results show visible difference in H3.3 signals peaking at CT4 and CT8. However, the definition of nucleosome positions on either side of the TSS is lost in this analysis. Therefore, we have included the panel shown below as a Supplementary Figure S2a.

3. Results for Fig. 2 should be clarified. Is the effect specific to CLOCK-BMAL1 targets, or for all genes. The authors' finding that H3.3 is enriched at both CLOCK:BMAL1 sites and CTCF sites suggest that the effects in PerKO are general, see comment 1 as well.

Our analysis shows a clear enrichment for H3.3 at CLOCK:BMAL1 sites (see response to comment 2 above) compared to non-clock genes. The enrichment we see at CTCF sites is largely at those sites in the proximity of CLOCK:BMAL1 sites or at PER2-BMAL1 sites (Figure 2c middle and right panels). H3.3 is enriched to a lesser extent globally at all CTCF sites in the PerKO liver. Thus, some of the changes in gene expression in PerKO animals are likely due to secondary effects from changes in CTCF sites.

4. Section on BRG1 and BRM ChIP-qPCR (Fig. 4). The higher recruitment of BRG1 at CT20 vs. CT8 seems opposite to expectations, yet not discuss. In fact, it seems that the recruitment of BRG1 and BRM are inversely correlated to H3.3 nucleosome signal. The authors should clarify this result, and whether it is in line with their models/hypotheses and/or the literature.

Indeed, the results of our BRG1 ChIP-qPCR experiments are different from what has been published (Kim JY *et al.*, 2014). The data from Kim *et al* in fact support our findings of BRG1 complexes assembling preferentially at CT8. We think that the differences could come from different antibodies used, or from the shorter crosslinking times used in our ChIP experiments.

We added the following text in the discussion:

Lines 388-396: "ChIP experiments between day and night for BRG1 did not correlate with the observed remodeler complex assembly data. Indeed, the results of our BRG1 ChIP qPCR experiments are different from what has been published (Kim JY et al., 2014), which support our findings that BRG1 complexes assemble preferentially at CT8. The differences could result from antibodies specificity, or from the shorter crosslinking times in our ChIP experiments. The discrepancy probably also arises from purification of soluble remodeler complexes that exist both on- and off-chromatin while ChIP experiments would enrich only for on-chromatin signals for the remodeler."

Based on our immunoprecipitation experiments, BRM should bind equally across all conditions. Indeed, we noted small differences but repeated experiments (n=5) show that these differences are not significant (Figure 4d).

5. Can the authors elaborate on why CLOCK:BMAL1 fail to bind fragile H3.3 nucleosomes in Per KO mice?

The reviewer raises an important point which indeed goes towards the earlier point raised- in that CLOCK:BMAL1 binding on chromatin in PerKO does not resemble daytime "activated" state.

There are two possible explanations that are not mutually exclusive.

1) A hypothesis we favor is rapid turnover of active CLOCK:BMAL1 complexes in PerKO tissues. This is also reflected in the BMAL1 binding data in Figure 5, where the overall open chromatin architecture in PerKO tissues would allow active CLOCK:BMAL1 to bind DNA directly, and for more rapid turnover on chromatin. This model also supports the general increase in transcription of clock-controlled genes in the PerKO liver.

Alternatively,

2) H3.3 nucleosomes at CT8 might be different from the H3.3 nucleosomes in PerKO tissues in PTMs landscapes or other histone variants within the nucleosome.

We have edited the text to reflect this point: Line 310-314.

"While H3.3 nucleosomes at CT8 could differ from those in PerKO livers in their PTM landscapes or the presence of other histone variants, the reduced association of BMAL1 with H3.3 complexes in PerKO chromatin (Fig. 3a) likely reflects BMAL1 directly binding E-boxes (Figure 5c) or an increased turnover at these sites."

Minor comments:

1. Lines 69-71: "Remarkably, H3.3A complexes retained H2A.Z but did not include canonical H3.1/2 at the two time points and in PerKO animals, CLOCK-BMAL1 was no longer enriched on variant H3.3 nucleosomes." I would describe the results in PerKO in a separate sentence, as the current reading is confusing (i.e., H3.1/2 is not included at the two time points and in PerKO animals, and CLOCK-BMAL1 was no longer enriched in H3.3 animals in PerKO).

We thank the referee for pointing this out and have changed the text (Lines 76-79) to clarify these thoughts.

Please, find a point-by-point response to the reviewer comments below.

Our responses are in blue.

Reviewer #1 (Remarks to the Author):

The authors have addressed all my concerns.

We thank the reviewer for taking their time in reviewing and evaluating our manuscript. We appreciate the suggestions, which helped us improve the quality of our manuscript.

Reviewer #2 (Remarks to the Author):

Authors have adequately answered reviewers' queries and this interesting article is now acceptable for publication.

We appreciate the referee for their thoughtful comments.

Reviewer #3 (Remarks to the Author):

The authors appropriately addressed my comments as well as those from the two other reviewers. I just note that a correction made by the authors led Fig. 2d to the referenced before other Fig. 2 panels, which needs to be corrected (see line 121). This manuscript is suitable for publication in Nature Communication.

We would like to thank the reviewer for taking their time in reviewing and evaluating our manuscript and helping us improve the interpretation of our data. We thank as well for the reviewer for pointing out the error regarding Figure 2d at line 121. This has now been fixed.